# HARE: Human-in-the-Loop Algorithmic Recourse

**Sai Srinivas Kancheti**[‡]                                                          *cs21resch01004@iith.ac.in*
*Indian Institute of Technology Hyderabad, India*
**Rahul Vigneswaran**[‡]                                                          *cs23mtech02002@iith.ac.in*
*Indian Institute of Technology Hyderabad, India*
**Bamdev Mishra**                                                                  *bamdevm@microsoft.com*
*Microsoft, India*
**Vineeth N Balasubramanian**                                                    *vineethnb@iith.ac.in*
*Indian Institute of Technology Hyderabad, India*

**Reviewed on OpenReview:** *https://openreview.net/forum?id=56EBglCFvx*

## Abstract

Machine learning models are seeing increasing use as decision making systems in domains such as education, finance and healthcare. It is desirable that these models are trustworthy to the end-user, by ensuring fairness, transparency and reliability of decisions. In this work, we consider a key aspect of responsible and transparent AI models – actionable explanations, *viz.* the ability of such models to provide recourse to end users adversely affected by their decisions. While algorithmic recourse has seen a variety of efforts in recent years, there have been very few efforts on exploring personalized recourse for a given user. Two users with the same feature profile may prefer vastly different recourses. The limited work in this direction hitherto rely on one-time feature preferences provided by a user. Instead, we present a human-in-the-loop formulation of algorithmic recourse that can incorporate both relative and absolute human feedback for a given test instance. We show that our formulation can extend any existing recourse generating method, enabling the generation of recourses that are satisfactory to the user. We perform experiments on 3 benchmark datasets on top of 6 popular baseline recourse methods where we observe that our framework performs significantly better on simulated user preferences.

## 1 Introduction

Machine Learning (ML) models are being increasingly utilized as decision-making tools in domains of significant human impact including recruitment, law, and education. Regulations and practical requirements have made model explanations a key requirement of modern ML systems for such application domains. While post-hoc model explanations provide a one-way view into the model's decision-making for a prediction, many applications can benefit from the availability of actionable explanations that allow a user to take specific actions to improve their outcome. *Algorithmic recourse* (Karimi et al., 2022; Verma et al., 2020) refers to the domain of methods that offer actionable recommendations to individuals who have been adversely affected by black-box decision making systems. The recommendations are usually offered as counterfactual explanations, which describe how a given user's feature profile should have appeared to be classified positively. Since the recommendations are meant to be implemented by the end user, we require the generated counterfactual explanations to be actionable and, ideally, to have a low execution cost. This is accomplished by answering the counterfactual question *"Given that the decision was y for input $x_f$, what would the decision be if the input had been $x_{cf}$ instead?"*. Here, $x_{cf}$ represents the counterfactual point, showing how the features would need to change from $x_f$ to yield the desired outcome. However, not all counterfactuals are useful—our aim is to generate realistic and feasible counterfactuals that individuals can effectively act upon.

Existing algorithms that generate recourses do so by solving a minimum cost optimization problem for each individual (Karimi et al., 2022). However, the notion of cost in many application domains can be subjective,

---

[‡]Equal Contribution

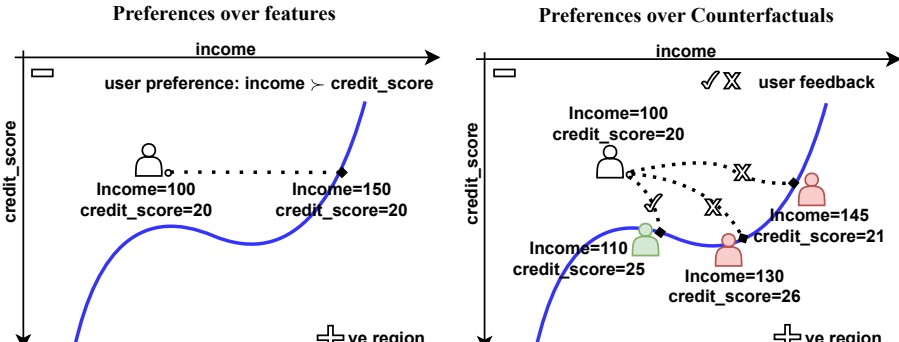

Figure 1: Personalized algorithmic recourse provides a pathway to consider individual user preferences in algorithmic recourse. The limited efforts so far Toni et al. (2024); Yetukuri et al. (2023) rely on feature-level user preferences. The above figure illustrates the benefits of instead considering user preferences over counterfactual samples. Consider the decision boundary (in blue) of a classifier over two features – *income* and *credit score*. The right subfigure shows three candidate recourses over which user feedback is taken (green indicates preferred recourse). Comparing plausible counterfactuals allows the user to make a holistic judgement. Left subfigure shows the consequence of providing feature preferences a priori (the user here prefers changing *income* over *credit score*). For this decision boundary, such a preference can admit a profile of $income = 150, credit\_score = 20$, which is clearly sub-optimal. Although the user may begin with an implicit preference for changing income, exposure to counterfactual choices reveals a more optimal preference.

with different individuals having differing notions of cost (for e.g., User A may incur a certain cost in increasing their credit score towards loan approval, while User B may have a different set of challenges or cost for a similar credit score increase). The difficulty of taking actions to modify their underlying features varies from person to person in a complex, possibly non-deterministic manner. Existing approaches tackle this problem by generating a set of diverse recourses giving greater choice to the user (Mothilal et al., 2019), or by asking individuals for preferences over specific features (Yetukuri et al., 2023; Singh et al., 2023a). We propose instead to view the process of generating a recourse as a *two-way interaction between the individual and the model*. An interactive formulation allows for finer preference elicitation, with simpler options for the user at each step, which elicits lower cognitive burden. Better knowledge of individual preferences allows us to find cheaper recourses for the specific individual, and discourages strategic behavior including gaming.

Many recourse generating algorithms use a fixed cost function for all individuals (Laugel et al., 2017; Poyiadzi et al., 2019; Pawelczyk et al., 2020; Antoran et al., 2021). This is not practical, since different individuals may have different notions of difficulty and actionability, which is not captured by a static cost function. To account for this, we propose to learn a metric for every individual based on their preferences. We present an iterative approach to interactively generate recourses. In each iteration, the individual is presented with a small set of *candidate recourses* and is queried about how suitable/unsuitable they are. This feedback is then used to learn a personalized distance metric for the individual. Based on this, a new set of recourses are generated which have a low cost according to the user's personalized metric. This iterative process is repeated until the user finds a satisfactory recourse among the generated candidates. In each iteration, we utilize preference information to learn a better metric for the individual.

Our approach differs from existing methods in a few ways: (i) compared to many existing methods Karimi et al. (2022) which do not consider a user's preference, we provide a two-way interactive approach for personalized algorithmic recourse; (ii) when compared to a few recent methods Toni et al. (2024); Yetukuri et al. (2023); Singh et al. (2023a) that consider such personalization of recourses, we elicit preferences over *counterfactuals* rather than features. While there has been a recent effort (Toni et al., 2024) that also allows interaction, they collect preferences over sequential interventions to update their estimate of the cost of interactions between features; we instead go beyond feature-wise preferences to directly query individuals on their preferred counterfactuals. As shown in Figure 1, this approach allows individuals to make judgments in the context of complete alternative outcomes, rather than isolated features changes. Instead of proposing directives to a user, we present a feasible counterfactual (or a set of them) and allow the user to decide

how they'd like to achieve the counterfactual based on their choice. We empirically study our proposed approach on 3 real-world datasets using 6 base recourse generation methods and 2 different machine learning models, and observe that it generates more personalized recourses than existing efforts. Our approach HARE (**H**uman-in-the-loop **A**lgorithmic **RE**course) is a simple, modular, and easily generalizable framework for generating personalized recourses. A key challenge in personalized recourse is that the form and nature of a user's cost function are often difficult to model explicitly. Unlike existing methods such as PEAR (Toni et al., 2024), which assume a structured cost function parametrized by a directed acyclic graph (DAG), HARE makes no assumptions about the underlying cost and instead learns user preferences directly from counterfactual choices. This approach provides greater control to the end user, allowing them to select their preferred recourse rather than relying on inferred intervention sequences. Furthermore, unlike PEAR, which is designed around a specific recourse generator (FARE (Toni et al., 2023)), HARE is fully modular and can be applied on top of any existing recourse generation method, making it applicable across different applications. A detailed comparison with PEAR is provided in Appendix § A. Considering the limited efforts so far, we also propose an evaluation scheme to measure user satisfaction by simulating a ground-truth recourse. Our experimental evaluation shows that HARE generates recourses that significantly outperform base recourse generation methods in terms of acceptance by end-users. Our code is publicly available[‡].

## 2 Related Work

**Algorithmic Recourse.** Methods to generate actionable counterfactual explanations, a problem called algorithmic recourse, have largely emerged in the last 4-5 years. A survey of existing methods has been presented in (Karimi et al., 2022; Verma et al., 2020). The most common family of methods find a recourse by minimizing a fixed distance function, also known as cost function, between the factual point and the recourse. These methods vary in the choice of cost functions employed as well as in the choice of constraints employed. Ustun et al. (2018) provided an Integer Programming formulation to find recourses for linear models. Wachter et al. (2017) proposed to use gradient descent to find recourses that minimize the Euclidean distance to the factual point. Other methods such as Laugel et al. (2017); Mothilal et al. (2019); Poyiadzi et al. (2019); Pawelczyk et al. (2020); Antoran et al. (2021) added other constraints such as diversity, closeness to the data manifold and classifier uncertainty. None of these methods however considered the preferences of the end user seeking recourse. Lack of alignment with a user's actionability or preference may lead to adverse outcomes such as users trying to game the system Hardt et al. (2015).

**Personalized Algorithmic Recourse.** While most initial recourse methods assumed a fixed cost function that is optimized for every individual, such methods cannot account for personal preferences and may lead to recourses which are not actionable or of low cost of the user. To address this, Mothilal et al. (2019) initially proposed to generate a set of diverse recourses to increase the probability that a desirable recourse is found for the user. However there is no guarantee that a preferred recourse belongs to the generated set. In more recent work, Yetukuri et al. (2023); Singh et al. (2023a) proposed to incorporate user preferences over features, either as a vector of numerical preference scores or as a ranking among features. While this offers a pathway to personalized recourse, eliciting feature-level preferences can place a high cognitive burden on a user, especially when a numerical score is required. Furthermore, these algorithms are susceptible to preference misspecification, and do not allow for post-hoc correction. HIP-CORE Abrate et al. (2024) is a recent human-in-the-loop approach that models user preferences using probability densities over counterfactuals. HIP-CORE solves a multi-object problem which aims to generate a set of recourses that satisfy various metrics such as sparsity, diversity and user preference. We differ from HIP-CORE in many important ways – i) HARE can build on top of any recourse generator baseline to generate personalized recourses; ii) HIP-CORE elicits a numerical user-preference for every counterfactual leading to a huge cognitive burden for the user (they ask over 300 numerical preferences per user). In contrast, we ask a small number of comparison queries (controlled via a budget $B = 30$), making our approach practical; iii) HIP-CORE models preferences via a separate density function for each feature, whose parameters are learned using preferences. In contrast, we work directly with the user's preferred counterfactual; iv) Actionability constraints cannot be encoded into HIP-CORE's optimization; and finally v) HIP-CORE does not compare against many baselines such as Wachter et al. (2017); Laugel et al. (2017); Poyiadzi et al. (2019). PEAR Toni et al. (2024) recently presented

---

[‡]https://github.com/rahulvigneswaran/HARE

an interactive approach to generate recourses, where the user selects from a set of feature-level *interventions* that maximize information gain. Different from these efforts, we instead propose a methodology to allow the user to select and work with counterfactuals directly, instead of dealing with features separately. In contrast with Toni et al. (2024), our approach does not require any modeling of the interactions between features, which may or may not align with the user's preferences. While we share the objectives of PEAR, our goal in proposing HARE is to have a simple, modular and easily generalizable approach for generating personalized recourses. It is often non-trivial to associate a form and nature to a user's cost function. HARE thus makes no assumptions on the user's cost, and relies only on the counterfactual choices made by the interacting user. HARE can work on top of any existing recourse generator, in contrast to PEAR which builds on top of FARE (Toni et al., 2023), and its empirical results provide strong evidence to show that such simple systems can work in practice. A detailed discussion of the differences between the two approaches in made in § A of the Appendix. We further discuss the benefits of presenting choices over direct counterfactuals in contrast to feature-level preferences in § 3.4.

**Learning from Human Feedback.** Preference elicitation methods aim to learn human preferences from data, and have had an independent history dating back to (Bradley & Terry, 1952). Traditional approaches leveraging feedback involve learning from logged data Gao et al. (2023); Ma et al. (2023), or assume a human choice oracle Mozannar et al. (2022). Algorithmic recourse is essentially intended to be an individualistic paradigm (Venkatasubramanian & Alfano, 2020), even more in high-stakes scenarios. Thus, for a new user, it is usually not possible to find data pertinent to the individual. Our proposed human-in-the-loop formulation allows for learning preferences over feasible recourses for each individual akin to having a dialog. It is well-known that asking users for quantitative preferences places a cognitive burden that can lead to noisy or incorrect choices Luce (1979); Keeney et al. (1977); Feffer et al. (2023). Our proposed algorithm thus utilizes *pairwise queries* Tesauro (1988); Yue et al. (2012); Joachims et al. (2007), where the user selects the best of two alternatives. We now present our methodology.

## 3 HARE: Methodology

In this section, we present HARE (**H**uman-in-the-loop **A**lgorithmic **RE**course), a framework to generate personalized recourses with a human-in-the-loop. Our framework is a way to generate personalized recourses for every user by interactively soliciting their preferences. Our human-in-the-loop formulation solicits human feedback over a set of exploratory recourses, which we term *candidate recourses*. Asking for preferences over a pair of recourses is more feasible, and allows the user to even consider interactions between features. This feedback is incorporated by the algorithm to generate refined candidates in the next iteration. The process ends when the user is satisfied with some recourse(s) from the candidate set. We describe each component of this proposed framework in the remainder of this section; a detailed algorithm of HARE is provided in § 3.4. HARE can work with any existing recourse-generating algorithm to integrate this human feedback component and generate personalized recourses. We begin by discussing the preliminaries.

### 3.1 Notations and Preliminaries

Consider an end user $u \in \mathcal{U}$ with a feature vector $x(u) \in \mathcal{X}$, and a fixed binary classifier $h : \mathcal{X} \to \{\pm 1\}$. We consider classifiers of the form $h(x) = sgn(f(x))$ where $sgn(\cdot)$ is the sign function and $f : \mathcal{X} \to \mathbb{R}$ is a scoring function. WLOG, we consider $+1$ the desired class, and $\mathcal{I} = \{x \in \mathcal{X} |\ h(x) = +1\}$ the region in feature space that is positively classified. We assume $x(u) \notin \mathcal{I}$ (since recourse is sought only when the desired class is not obtained as model prediction). Let $\mathcal{A}(u) \subseteq \mathcal{X}$ denote the set of feature values that the user can attain by making *actionable* changes to their feature vector. A recourse generator $\mathcal{M} : \mathcal{X} \to \mathcal{I}_{\text{fin}}(\mathcal{X})$, where $\mathcal{I}_{\text{fin}}(\mathcal{X})$ is the set of all finite subsets of $\mathcal{X}$, maps a user's feature vector to a finite set of recourses. The generated recourses are considered as valid counterfactuals and as being actionable by the user, i.e $\forall r \in \mathcal{M}(x(u)),\ h(r) = +1$ and $r \in \mathcal{A}(u)$. Our framework HARE can be integrated into any existing recourse generation method $\mathcal{M}$ to generate personalized recourses. We describe a few salient recourse generation methods in Appendix § C. HARE uses $\mathcal{M}$ to find a initial baseline recourse(s), which is used to sample exploratory candidate recourses, as described below.

### 3.2 Generation of Candidate Recourses

To elicit preferences of a user $u$, we require a set of recourses over which feedback can be provided. To this end, we sample candidate recourses around a baseline recourse $r_0(u)$, which is generated using any existing recourse generation method. We present an algorithm *ActionableSampling* to sample $k$ actionable candidate recourses around a baseline recourse $r_0(u)$ using gradient descent. The set of personalized candidate recourses $\mathcal{C}(u)$ should be diverse enough to capture possible variations in a user's preference, but not too large as to be cognitively burdensome to query the user. The output of *ActionableSampling* is hence a set of diverse candidate recourses which are valid w.r.t user $u$ and base classifier $g$, i.e. they are actionable for $u$ and are classified positively by $g$.

We now herein describe Algorithm 1. We start with $k$ random actionable exploratory direction vectors $\{p_1, \ldots, p_k\}$, $p_i \in \mathcal{X}$, which are added to $r_0$ to generate the candidate recourses. Here, $k$ is decided by the query budget of the user, as discussed in § 3.4. We require the exploratory directions to be as diverse as possible, while remaining actionable to the user. To make the directions as diverse as possible, we follow Mettes et al. (2019) and minimize the mean maximum similarity between the directions. Let $P = [p_1, \ldots, p_k]$, $P \in \mathbb{R}^{|\mathcal{X}| \times k}$ be the matrix with directions as columns. The similarity matrix is $P^\mathsf{T} \cdot P - 2 \cdot \mathbb{I}$, where we subtract the identity to remove self-similarity. Denoting similarity loss by $L_1(P)$, we have:

---

**Algorithm 1:** ActionableSampling

**Input** : Baseline recourse $r_0(u)$, Classifier $g$, Number of candidates $k$, Actionability set $\mathcal{A}(u)$, Gradient descent iterations $n$, Hyp. $\lambda$

**Output :** Candidate recourses $\mathcal{C}(u)$

**Initialize:** $k$ random actionable directions $P_1 = \{p_1, \ldots, p_k\}$ such that $p_i \in \mathcal{A}(u) \forall i$

1 **for** $i = 1, 2, \ldots, n$ **do**
2 $\quad L_1(P_i) \leftarrow mean(row\_max(P_i \cdot P_i^\mathsf{T} - 2 \cdot \mathbb{I}))$
3 $\quad L_2(P_i) = BinaryCrossEntropy(g(r_0(u) + \gamma \cdot P_i), 1)$
4 $\quad P_i' \leftarrow P_i - \eta \cdot \nabla (\lambda L_1(P_i) + L2(P_i))$
5 $\quad P_{i+1} \leftarrow \{\underset{p \in \mathcal{A}(u)}{\operatorname{argmin}} \|p - p_j\| \mid p_j \in P_i'\}$
6 $\quad P_{i+1} \leftarrow \{\frac{p}{\|p\|} \mid p \in P_{i+1}\}$
7 $P^\star(u) \leftarrow P_{i+1}$
8 $\mathcal{C}(u) \leftarrow \{r_0(u) + \gamma \cdot p \mid p \in P^\star(u)\}$
9 **return** $\mathcal{C}(u)$

---

$$L_1(P) = \frac{1}{k} \sum_{i \in [k]} \max_{j \in [k]} (P^\mathsf{T} \cdot P - 2 \cdot \mathbb{I})_{ij} \tag{1}$$

Minimizing Eq 1 generates directions that are away from each other, and hence diverse. The candidate recourses are generated from the exploratory directions as $r_0(u) + \gamma \cdot P$, where $\gamma$ is a magnitude hyperparameter. To ensure that sampled candidate recourses are valid w.r.t to the base classifier $g$, we add a binary cross entropy loss $L_2(P)$:

$$L_2(P) = BinaryCrossEntropy(g(r_0(u) + \gamma \cdot P), 1) \tag{2}$$

Minimizing this loss ensures that the base classifier assigns a label of $+1$ to the candidate recourses. Subsequently, as shown in Algorithm 1, we perform $n$ steps of Projected Gradient descent over $\lambda \cdot L_1(P) + L_2(P)$ to yield valid, diverse directions $P^\star$. After each gradient descent step, actionability is ensured by projecting the directions onto the set $\mathcal{A}(u)^\ddagger$. The output set of candidate counterfactuals is given by $\mathcal{C}(u) = \{r_0(u) + \gamma \cdot p \mid p \in P^\star(u)\}$.

In the candidate generation process, features identified as immutable for a user (when such information is available) remain unchanged, ensuring all candidates are actionable. While our method does not rely on immutability inputs from users, it can seamlessly incorporate them when provided.

#### 3.2.1 Calibrating Candidate Recourses

While a set of candidate recourses are now available, it is important that these candidate recourses are also those of minimal cost to the end user. To achieve this, we calibrate each candidate recourse $c$ by identifying a sample that is in the direction of $c$, but closest to the decision boundary from $x(u)$.

---

$\ddagger$We abuse notation slightly and use $\mathcal{A}(u)$ to also indicate the set of actionable feature *changes*.

We treat this as a root-finding problem $f(x) = 0$ for the classifier scoring function $f$ which captures the point on the line joining $x(u)$ and $c$ that is closest to the decision boundary.

Since $f(x(u)) < 0$ and $f(c) > 0$ (by construction of recourse methods), there is at least one root in $[x(u), c]$. For classifiers with non-linear decision boundaries, we are interested in the root closest to $c$. We use a modified version of the binary search algorithm for this purpose, which we call *BoundaryPointSearch*. This variant is summarized in Algorithm 2. Algorithm 3 describes the procedure to generate the final candidate recourses $\mathcal{C}'(u)$ by using Binary Search on every candidate. The user $u$ can now indicate their preferences over $\mathcal{C}'(u)$, which we describe next.

---

**Algorithm 2:** BoundaryPointSearch

**Input** : User feature $x(u)$, Candidate recourse $c$, Classifier $g$, Tolerance $\epsilon$

**Output** : Recourse on decision boundary $c'$

**Initialize:** $start \leftarrow x(u)$, $end \leftarrow c$

1 **while** $\|start - end\|_2 \geq \epsilon$ **do**
2     $mid \leftarrow \dfrac{start + end}{2}$
3     **if** $g(mid) \geq 0.5$ **then**
4        $\lfloor$ $end \leftarrow mid$
5     **else**
6        $\lfloor$ $start \leftarrow mid$

7 $c' \leftarrow end$
8 **return** $c'$

---

### 3.3 Eliciting Feedback over Candidate Recourses

Existing works that consider user feedback in algorithmic recourse (Yetukuri et al., 2023; Toni et al., 2024) require preferences over individual features, where the user either provides a numerical score indicating their preference for changing a feature, or provide a ordering over features. Eliciting such feature-wise preferences may be non-trivial or even infeasible for a user due to multiple reasons: (i) Providing feature-wise preferences place a high cognitive burden on the user, especially for algorithms that require a numerical value of features importance or applications where there are a large number of features; (ii) It may be difficult for a user to fully order features, since ordering implicitly involves $\mathcal{O}(d^2)$ pairwise comparisons for $d$-dimensional data; (iii) It is also possible that there may be many feasible recourses that satisfy such feature ranking preferences, and the user may want to choose between them; (iv) Care has

---

**Algorithm 3:** FinalCandidateRecourses

**Input** : User feature $x(u)$, Classifier $g$, Initial candidate recourses $\mathcal{C}(u)$, Tolerance $\epsilon$

**Output** : Final candidate recourses $\mathcal{C}'(u)$

**Initialize:** $\mathcal{C}'(u) \leftarrow \{\}$

1 **for** $c \in \mathcal{C}(u)$ **do**
2     $c' \leftarrow$ BoundaryPointSearch$(x(u), c, g, \epsilon)$
3     $\mathcal{C}'(u) \leftarrow \mathcal{C}'(u) \cup \{c'\}$

4 **return** $\mathcal{C}'(u)$

---

to be taken to ensure that the algorithm is robust to preference misspecification, which again scales with dimensionality. We thus propose an alternate mechanism, where users simply declare their preferences over candidate recourses themselves. We consider two types of feedback provided by a user $u$ over candidates from $\mathcal{C}'(u)$:

**Relative Feedback:** The user compares a pair of recourses and chooses their preferred recourse. Relative feedback is modeled with a Comparative Evaluation ($CE$) function over a pair of recourses $r_i$ and $r_j$.
**Absolute Feedback:** Given a counterfactual, the user either accepts it or rejects it. Absolute feedback is modeled with an Indicator Evaluation ($IE$) given recourse $r$.

$$CE(r_i, r_j) = \begin{cases} +1 & r_i \succ_u r_j \\ 0 & r_i \prec_u r_j \end{cases} \qquad IE(r) = \begin{cases} +1 & r \text{ is satisfactory} \\ -1 & \text{otherwise} \end{cases}$$

Relative feedback is easier to provide due to its lower cognitive load and reduced chances of the user making an erroneous judgement Tesauro (1988); Yue et al. (2012). We hence primarily rely on $CE$ queries to solicit feedback over the set of candidate recourses. We use absolute feedback as a termination criterion, when the user has been exposed to multiple comparisons, and is ready to make a final decision.

### 3.4 The Overall HARE Framework

Given a calibrated candidate set $\mathcal{C}'(u)$, we allow for asking relative and absolute feedback queries to the user $u$ in our overall framework to return personalized recourses. For instance, we begin by asking a $CE$ (or $IE$) over random pair of recourses from $\mathcal{C}'(u)$. The winner is then compared to an un-visited candidate recourse,

and the process repeats till the best candidate recourse $r^\star$ is found. The above process is repeated over iterations, where the $r^\star$ at the end of an iteration $t-1$ is used as the base recourse to compute the candidate recourses in iteration $t$. Additionally, our approach allows us to provide the user with their preferred recourses within a fixed query budget. Assuming each query ($CE$ or $IE$) consumes one unit of query budget and an overall user budget $B$, we allow $\frac{B}{T}$ candidate recourses in each of $T$ iterations of the overall process to stay within the specified budget.

Algorithm 4 summarizes our framework's methodology. Without loss of generality, we consider the starting point of our algorithm as a user $u$ with a data sample $x(u)$ that has been unfavorably (negatively) classified by a classifier $g$ (i.e. an undesirable prediction, which the user seeks to reverse).

---

**Algorithm 4:** HARE

    **input**    **:** User feature $x(u)$, actionability set $\mathcal{A}(u)$, classifier $g$, Base Recourse Generator $\mathcal{M}$, user relative and absolute feedback functions $CE_u$ & $IE_u$, user query budget $B$, total iterations $T$, gradient descent iterations $n$, regularization hyperparam $\lambda$, tolerance $\epsilon$

    **output**  **:** Preferred recourse $r^\star(u)$

    **initialize:** $r_0(u) \leftarrow \mathcal{M}(x(u))$

**1** **for** $t = 1, 2, \ldots, T$ **do**

**2**     $\mathcal{C}_t(u) \leftarrow \text{ActionableSampling}(\text{baseline\_recourse} = r_{t-1}(u), \text{classifier} = g, \text{number\_of\_candidates} =$
    $\frac{B}{T}, \text{Actionability\_set} = \mathcal{A}(u), \text{gd\_iterations} = n, \text{regularization\_param} = \lambda)$

**3**     $\mathcal{C}'_t(u) \leftarrow \text{FinalCandidateRecourses}(\text{user\_feature} = x(u), \text{classifier} = g, \text{candidate\_recourses} =$
    $\mathcal{C}_t(u), \text{tolerance} = \epsilon)$

**4**     **while** $|\mathcal{C}'_t(u)| > 1$ **do**

**5**         pick $r_i, r_j$ from $\mathcal{C}'_t(u)$

**6**         $\mathcal{C}'_t(u) \leftarrow \mathcal{C}'_t(u) \setminus \{CE_u(r_i, r_j) \cdot r_j + (1 - CE_u(r_i, r_j)) \cdot r_i\}$

**7**     $r_t(u) \leftarrow \mathcal{C}'_t(u)$

**8**     **if** $IE_u(r_t(u)) = 1$ **then**

**9**         **terminate loop**

**10** $r^\star(u) \leftarrow r_t(u)$

**11** **return** $r^\star(u)$

---

For each iteration $t \in \{1, \ldots, T\}$, the following steps are performed:

1. HARE first generates a baseline recourse $r_{t-1}(u)$ using any off-the-shelf recourse generator $\mathcal{M}$ (or) as the best candidate returned by the previous iteration

2. In Line 2 of Algorithm 4, $\frac{B}{T}$ diverse initial candidate recourses $\mathcal{C}_t(u)$ are generated around $r_{t-1}(u)$ via the ActionableSampling algorithm described in § 3.2

3. In Line 3, these initial candidates are calibrated to obtain the final candidate recourses $\mathcal{C}'_t(u)$ that are relatively closer to the decision boundary of $g$, via the FinalCandidateRecourses algorithm 3

4. In Lines 4-6, as described in § 3.3, the best candidate $r_t^\star(u)$ is chosen over $\mathcal{C}'_t(u)$ by making $B$ comparative evaluations.

5. In Line 7, the best candidate is chosen as the baseline recourse for the next iteration

The overall algorithm has a query budget of $B$. The algorithm terminates when the query budget is exhausted or when the user is happy with the generated recourse (Line 8 of Algorithm 4). In practice, there must be a two-way dialog between the user and the recourse generator. If the user is unable to accept any of the candidates after many rounds of feedback, the recourse generator must be able to accept suggestions from the user and check if they are valid. For instance, consider the example from Figure 1 of the main paper where the user is provided the final 3 options (income, credit_score) of (110,25), (130,26), (145,21). Even if the user is unable to accept any of these, her exposure to the choices allows her to suggest a possible recourse that works. For instance, she may state that (100,25) is doable, and in this example is a valid recourse. Note that the user may have an apriori implicit preference of changing income over credit score (perhaps

they want to be future proof), but the exposure to counterfactual choices (which are constrained by the underlying predictive model) can reveal different preferences. Furthermore, HARE is flexible, allowing end users to dictate the kinds of choices available to them. For example, the ActionableSampling algorithm 1 ensures that candidates are diverse. It also allows users to specify new actionability constraints on the fly, and to restrict the range of sampling by tweaking $\gamma$. For consequential decision making systems (loans, jobs, crime etc), it is in the end user's best interest to achieve a recourse that they can act on, and our method presents a simple and modular framework to the end user, where the user preference can be integrated with any existing recourse generation method.

## 4 Experiments, Results and Analysis

We comprehensively evaluate HARE, both with number of iterations $T = 1$ and $T > 1$, on 3 real-world benchmark datasets commonly used in earlier similar work. For convenience of reading, we refer to the use of our method when $T = 1$ as HARE and when $T > 1$ as Multi-HARE. As described earlier, our framework generates personalized recourses by incorporating a human-in-the-loop, and can be applied on top of any base recourse generator. Therefore, we study the effectiveness of our framework across 6 diverse base recourse generation methods. Since recourse can be applied on top of any machine learning model, we perform experiments with two kinds of classifiers: a 3-layer Artificial Neural Network (ANN) and Logistic Regression (LR). Our results demonstrate that HARE generates more personalized recourses compared to any standalone base recourse generator.

### 4.1 Datasets, Simulating Ground-Truth Recourses & Evaluation Metrics

**Datasets.** We evaluate on 3 commonly used binary datasets spanning different application domains including credit worthiness, criminal recidivism, and income prediction, which are popularly used in recourse literature. `Adult Income` Becker & Kohavi (1996) is a binary classification dataset with 13 features (6 continuous, 7 categorical) that are used to predict if annual income exceeds $50K$. We binarize the categorical features, and consider *'age', 'sex', 'fnlwgt', 'native-country', 'race', 'relationship'* to be immutable features in this work. `GiveMeSomeCredit` Kaggle (2021) is used to predict credit worthiness and consists of 10 continuous features, out of which we consider *'age', 'NumberOfDependents'* to be immutable. Finally we consider `COMPAS` Larson et al. (2016), which is used to predict recidivism risk. It consists of 7 features (4 continous and 3 categorical) out of which *'age', 'race', 'sex'* are considered immutable. For all datasets, continuous features are scaled to lie within the range $[0, 1]$ and categorical features are binarized.

**Base Recourse Generators.** HARE can be integrated with any existing recourse generation algorithm to generate personalized recourses. We evaluate its performance on 6 popular baseline recourse methods: Wachter Wachter et al. (2017), Growing Spheres (GS) Laugel et al. (2017), FACE Poyiadzi et al. (2019), CCHVAE Pawelczyk et al. (2020), CLUE Antoran et al. (2021) and CRUDS Downs et al. (2020). These baselines represent a diverse collection of recourse generators which differ widely in their scope and approach. A short description of these baseline methods is given in Appendix §C for completeness.

**Simulating Ground-Truth Recourses.** Since real human feedback is unavailable, we simulate a ground-truth recourse to approximate a user's preferences. This ground-truth recourse serves two purposes: (i) It provides a reference for simulating user feedback – for instance, relative feedback on two candidate recourses is simulated by comparing each of them against the ground-truth, and (ii) It allows us to quantitatively evaluate user satisfaction of a generated recourse. Given a user $u$ with feature profile $x(u)$, the g.t. recourse $r^{gt}(u)$ captures the recourse most preferred by the user. Importantly, this preference is not known to the algorithm. We simulate $r^{gt}(u)$ for $u$ by randomly sampling actionable recourses on the classifier decision boundary. We sample $r^{gt}$ at three scales – *near*, where the g.t recourse is close in euclidean distance to the factual point; *intermediate* and *far*. This categorization allows us to model users with different preferences – some may prefer small adjustments across multiple features (*near*), while others may prefer changing a single feature by a large amount. Additionally, users at the far scale may also represent those willing to incur higher costs for a more robust recourse that remains valid under classifier shifts Upadhyay et al. (2021). We evaluate the performance of personalized recourse methods by measuring how closely the generated recourses align with simulated ground-truths. Note that the only information HARE gets about $r^{gt}(u)$ is obtained

indirectly, through simulated user feedback. Standalone base recourse generators do not incorporate user feedback and have no way of obtaining information about the g.t recourse.

**Evaluation Metrics:** We propose two new metrics to measure end user satisfaction with personalized recourses – (i) *Ground-Truth Proximity (GTP)* measures the L2 (Euclidean) distance between the generated recourse and the simulated g.t recourse. A low GTP value indicates that the generated recourse is close to the user's ideal solution, and ii)*Ground-Truth Dissimilarity (GTD)* measures the cosine distance between the vectors from the factual point to the generated recourse and to the g.t recourse. A low value of GTD implies that the generated recourse aligns in direction with the user's preferred changes. These new metrics specifically assess how well the generated recourse matches a user's preference.

We also evaluate on classical recourse metrics such as *Success Rate*, which measures the fraction of test individuals for which a recourse could be generated; *Constraint Violations*, which measures the average number of immutable features changed; *Redundancy*, which measures the average number of unnecessary feature changes; *Proximity*, the L2 distance between the factual point and generated recourse; and *Sparsity*, which measures the number of features modified to generate the recourse. For all metrics except Success rate, lower is better.

## 4.2 Implementation Details

**Classifier.** We train and fix a 3-layered Artifical Neural Network (ANN) and a Logistic Regression (LR) model for each dataset.Recourses are generated for 150 fixed individual samples taken from the test-set.
**Recourse Generator Baselines.** We use recourse generator implementations provided by CARLA Pawelczyk et al. (2021) for all 6 baselines: Wachter, GS, FACE, CCHVAE, CLUE and CRUDS. Hyperparameters for each baseline are set to values that maximize success rate of recourse generation, shown in Appendix § D.
**HARE.** We have a total budget of $B = 30$ user queries (across iterations) unless specified otherwise (i.e. if $T = 5$, we allow for 6 queries per iteration in this setting). Since the end user is often motivated to achieve a positive classification, we believe that this query budget is reasonable. However, for completeness of understanding, we study the efficacy of personalized recourse generation under different user budgets in § 4.4. The iterative variant of our approach, termed Multi-HARE, divides the total query budget into multiple iterations, allowing for iterative refinement of the recourse based on user feedback. Unless specified, we choose $T = 5$ iterations. For *ActionableSampling*, we perform full-batch gradient descent using the Adam Kingma & Ba (2014) optimizer for $n = 100$ iterations with a learning-rate of 0.1. We set the magnitude hyperparameter $\gamma$ to 1, and the regularization hyperparameter $\lambda$ to 10. In *BoundaryPointSearch* the tolerance value $\epsilon$ is set to $1e-06$. All experimental results are averaged over 5 seeds to ensure robustness. Our code will be made publicly available upon acceptance.

## 4.3 Results

We now present the results of our experiments, where we evaluate HARE across a variety of settings. Our goal is to assess how effectively HARE generates personalized recourses compared to standalone base recourse generators. We study five keys aspects – i) *Measuring User Satisfaction*, where we measure GTP & GTD defined in § 4.1, to evaluate the degree to which generated recourses align with the simulated ground-truth recourses; ii) *Budget Ablation*, where we examine how varying the number of user queries affects the quality of generated recourses; iii) *Multi-HARE Ablation*, where we study the impact of iterative refinement on recourse generation; iv) *BoundaryPointSearch Ablation*, where we evaluate the effect of not performing a *BoundaryPointSearch* for every candidate recourse; and v) *Ground-Truth Recourse Scale Ablation*, where we investigate the performance of HARE across three different scales of ground-truth recourses. Finally we report classical recourse metrics as discussed in § 4.1. These metrics do not measure the degree of personalization of the generated recourses, but we report them for completeness. These experiments help us understand the various components of HARE and study how it performs under varying conditions.

**Measuring User Satisfaction.** In Table 1, we show the results for all six baseline recourse generators across three datasets, evaluated on both ANN and LR classifiers. For each baseline, we find that HARE and its iterative variant, Multi-HARE, consistently obtain recourses that align more closely with the simulated ground-truths as measured by both Ground-Truth Proximity (GTP) and Ground-Truth Dissimilarity (GTD).

| | Base Recourse | Ours | ADULT | | COMPAS | | CREDIT | |
|---|---|---|---|---|---|---|---|---|
| | | | GTP ↓ | GTD ↓ | GTP ↓ | GTD ↓ | GTP ↓ | GTD ↓ |
| ANN | Wachter | Base | 0.93±0.38 | 0.87±0.03 | 0.39±0.34 | 0.75±0.30 | 1.07±0.18 | 0.97±0.26 |
| | | + Random | 1.01 ±0.35 | 1.17 ±0.14 | 0.39 ±0.24 | 0.87 ±0.27 | 1.21 ±0.15 | 1.09 ±0.1 |
| | | + HARE | 0.69±0.31 | 0.09±0.04 | **0.27**±0.26 | 0.27±0.17 | **0.89**±0.17 | **0.37**±0.09 |
| | | + Multi-HARE | **0.58**±0.28 | **0.08**±0.03 | **0.27**±0.26 | **0.26**±0.15 | 0.90±0.22 | 0.38±0.12 |
| | GS | Base | 0.92±0.39 | 0.82±0.09 | 0.40±0.31 | 0.80±0.14 | 1.06±0.19 | 1.06±0.09 |
| | | + Random | 0.95 ±0.35 | 0.97 ±0.35 | 0.47 ±0.29 | 0.99 ±0.24 | 1.23 ±0.14 | 1.2 ±0.18 |
| | | + HARE | 0.71±0.34 | 0.10±0.05 | 0.26±0.22 | **0.21**±0.10 | 0.89±0.18 | 0.40±0.06 |
| | | + Multi-HARE | **0.56**±0.27 | **0.08**±0.03 | **0.25**±0.23 | **0.21**±0.10 | **0.83**±0.21 | **0.30**±0.07 |
| | FACE | Base | 1.06±0.36 | 1.06±0.09 | 0.42±0.32 | 0.82±0.16 | 1.11±0.19 | 1.09±0.17 |
| | | + Random | 1.04 ±0.39 | 1.43 ±0.14 | 0.44 ±0.30 | 0.88 ±0.15 | 1.21 ±0.13 | 1.09 ±0.05 |
| | | + HARE | 0.77±0.35 | 0.24±0.05 | 0.26±0.22 | **0.21**±0.10 | 0.90±0.18 | 0.42±0.09 |
| | | + Multi-HARE | **0.56**±0.26 | **0.08**±0.02 | **0.25**±0.24 | **0.21**±0.10 | **0.83**±0.22 | **0.32**±0.08 |
| | CCHVAE | Base | 1.32±0.25 | 0.99±0.03 | 0.42±0.32 | 0.76±0.24 | 1.13±0.20 | 1.01±0.13 |
| | | + Random | 1.10 ±0.31 | 1.29 ±0.06 | 0.45 ±0.30 | 0.91 ±0.17 | 1.20 ±0.14 | 1.03 ±0.05 |
| | | + HARE | 0.83±0.35 | 0.48±0.06 | **0.29**±0.30 | 0.26±0.11 | 0.91±0.18 | 0.44±0.07 |
| | | + Multi-HARE | **0.60**±0.29 | **0.07**±0.03 | **0.29**±0.31 | **0.25**±0.14 | **0.83**±0.21 | **0.32**±0.07 |
| | CLUE | Base | - | - | 0.95±0.19 | 0.71±0.19 | 1.22±0.18 | 1.04±0.10 |
| | | + Random | - | - | 0.47 ±0.29 | 0.76 ±0.23 | 1.19 ±0.14 | 1.06 ±0.1 |
| | | + HARE | - | - | 0.32±0.26 | 0.33±0.12 | 0.93±0.17 | 0.49±0.09 |
| | | + Multi-HARE | - | - | **0.26**±0.24 | **0.23**±0.10 | **0.84**±0.21 | **0.33**±0.07 |
| | CRUDS | Base | 2.16±0.54 | 1.12±0.12 | 1.22±0.15 | 0.70±0.20 | 1.20±0.22 | 1.01±0.16 |
| | | + Random | 1.26 ±0.23 | 1.20 ±0.11 | 0.52 ±0.36 | 0.78 ±0.22 | 1.16 ±0.13 | 0.84 ±0.12 |
| | | + HARE | 1.09±0.36 | 1.00±0.14 | 0.23±0.05 | 0.38±0.09 | 0.77±0.19 | 0.34±0.08 |
| | | + Multi-HARE | **0.50**±0.19 | **0.09**±0.02 | **0.15**±0.03 | **0.15**±0.02 | **0.64**±0.15 | **0.23**±0.04 |
| LR | Wachter | Base | 0.94±0.35 | 0.80±0.10 | 0.25±0.06 | 0.58±0.17 | 0.90±0.18 | 0.73±0.05 |
| | | + Random | 1.04 ±0.31 | 1.10 ±0.11 | 0.34 ±0.14 | 0.88 ±0.38 | 0.95 ±0.14 | 0.86 ±0.10 |
| | | + HARE | 0.63±0.27 | 0.10±0.01 | 0.16±0.05 | 0.14±0.07 | 0.72±0.17 | 0.30±0.06 |
| | | + Multi-HARE | **0.39**±0.17 | **0.07**±0.02 | **0.15**±0.04 | **0.13**±0.08 | **0.65**±0.15 | **0.21**±0.02 |
| | GS | Base | 0.91±0.38 | 0.72±0.17 | 0.28±0.08 | 0.89±0.18 | 0.93±0.17 | 0.87±0.06 |
| | | + Random | 0.91 ±0.32 | 0.72 ±0.26 | 0.32 ±0.15 | 0.91 ±0.42 | 0.98 ±0.27 | 0.87 ±0.26 |
| | | + HARE | 0.64±0.29 | 0.12±0.04 | 0.17±0.05 | **0.15**±0.07 | 0.73±0.16 | 0.29±0.01 |
| | | + Multi-HARE | **0.39**±0.21 | **0.05**±0.02 | **0.16**±0.06 | 0.17±0.13 | **0.63**±0.18 | **0.19**±0.06 |
| | Face | Base | 1.19±0.31 | 1.03±0.12 | 0.27±0.07 | 0.72±0.15 | 0.96±0.21 | 0.91±0.10 |
| | | + Random | 1.13 ±0.33 | 1.06 ±0.11 | 0.31 ±0.14 | 0.62 ±0.16 | 0.95 ±0.26 | 0.85 ±0.29 |
| | | + HARE | 0.77±0.32 | 0.30±0.06 | 0.15±0.05 | **0.14**±0.07 | 0.74±0.19 | 0.31±0.04 |
| | | + Multi-HARE | **0.43**±0.20 | **0.07**±0.02 | **0.15**±0.05 | 0.16±0.12 | **0.63**±0.18 | **0.19**±0.07 |
| | CCHVAE | Base | 1.42±0.20 | 0.96±0.05 | 0.31±0.09 | 0.75±0.22 | 0.98±0.24 | 0.86±0.16 |
| | | + Random | 1.22 ±0.30 | 0.99 ±0.02 | 0.27 ±0.08 | 0.65 ±0.17 | 0.94 ±0.28 | 0.81 ±0.31 |
| | | + HARE | 0.87±0.30 | 0.50±0.03 | 0.12±0.02 | **0.12**±0.07 | 0.74±0.21 | 0.30±0.06 |
| | | + Multi-HARE | **0.50**±0.23 | **0.09**±0.04 | **0.13**±0.04 | 0.13±0.04 | **0.63**±0.19 | **0.19**±0.07 |
| | CLUE | Base | - | - | 0.89±0.03 | 0.67±0.13 | 1.07±0.23 | 0.87±0.13 |
| | | + Random | - | - | 0.27 ±0.10 | 0.68 ±0.30 | 0.94 ±0.27 | 0.80 ±0.29 |
| | | + HARE | - | - | 0.20±0.05 | 0.27±0.09 | 0.76±0.20 | 0.33±0.08 |
| | | + Multi-HARE | - | - | **0.16**±0.05 | **0.17**±0.13 | **0.64**±0.18 | **0.20**±0.07 |
| | CRUDS | Base | 2.92±0.45 | 1.22±0.12 | 1.27±0.02 | 0.70±0.11 | 1.04±0.23 | 0.85±0.14 |
| | | + Random | 1.60 ±0.46 | 1.17 ±0.13 | 0.30 ±0.13 | 0.75 ±0.23 | 0.93 ±0.21 | 0.81 ±0.16 |
| | | + HARE | 1.09±0.36 | 1.00±0.14 | 0.23±0.05 | 0.38±0.09 | 0.77±0.19 | 0.34±0.08 |
| | | + Multi-HARE | **0.50**±0.19 | **0.09**±0.02 | **0.15**±0.03 | **0.15**±0.02 | **0.64**±0.15 | **0.23**±0.04 |

Table 1: Results on Measuring User Satisfaction for the *far* g.t recourse scale. We show results on three datasets (in columns), and six baseline recourse generators (in rows) for two classifiers, ANN & LR. We do not report numbers for CLUE on Adult as the success rate is too. low For each baseline, we show the impact of incorporating user feedback using HARE and Multi-HARE. Results in **bold** indicate best numbers, and underline second-best. We observe that our framework can effectively incorporate user feedback.

We achieve lower GTP and GTD scores, indicating that the generated recourses are both spatially closer to the user's ideal solution and also align better in terms of direction of change. For both methods, we use a total user query budget of $B = 30$ queries, while Multi-HARE performs recourse refinement by distributing the queries across $T = 5$ iterations. Multi-HARE generates the most personalized recourses, suggesting that refining recourses across multiple iterations can better capture user preferences. Moreover it presents a smaller candidate set at each iteration, making it easier for users to express preferences at each stage, leading to lower cognitive load overall. In the random baseline, we pick a candidate at random from the final candidate set at every iteration, instead of using the feedback provided by the user. We add this simple baseline to show that actually using user feedback is crucial to obtain good quality recourses.

### 4.4 Analysis and Ablation Studies

**Study of Budget Choice.** In real-world scenarios, increased interaction with end-users typically leads to better personalized recourses. To explore the sensitivity of HARE to the query budget, we conduct an ablation study with different budget allocations. The results, shown in Table 2, demonstrate that there is a consistent improvement in both GTP and GTD metrics as the budget increases. Notably, the performance gains are significant up to a budget of 30 queries, beyond which improvements plateau. While larger budgets yield more personalized recourses, a budget of around 30-70 queries is optimal and strikes a balance between user satisfaction and cognitive load. In § B.3 in the Appendix, we study the performance of HARE under even lower budgets.

| | Base Recourse | Budget (B) | ADULT GTP ↓ | ADULT GTD ↓ | COMPAS GTP ↓ | COMPAS GTD ↓ | CREDIT GTP ↓ | CREDIT GTD ↓ |
|---|---|---|---|---|---|---|---|---|
| ANN | Wachter | Base | 0.93±0.38 | 0.87±0.03 | 0.39±0.34 | 0.75±0.30 | 1.07±0.18 | 0.97±0.26 |
| | | 10 | 0.69±0.31 | 0.10±0.04 | 0.33±0.34 | 0.40±0.24 | 0.95±0.18 | 0.42±0.14 |
| | | 30* | 0.69±0.31 | **0.09**±0.04 | 0.27±0.26 | 0.27±0.17 | 0.89±0.17 | 0.37±0.09 |
| | | 70 | **0.67**±0.33 | 0.10±0.03 | **0.24**±0.23 | 0.21±0.16 | 0.86±0.18 | **0.32**±0.04 |
| | | 100 | **0.67**±0.33 | 0.12±0.03 | **0.24**±0.25 | **0.20**±0.14 | **0.85**±0.19 | 0.33±0.04 |
| | GS | Base | 0.92±0.39 | 0.82±0.09 | 0.40±0.31 | 0.80±0.10 | 1.06±0.19 | 1.06±0.09 |
| | | 10 | 0.75±0.34 | 0.23±0.07 | 0.34±0.31 | 0.38±0.09 | 0.91±0.15 | 0.45±0.16 |
| | | 30* | 0.71±0.34 | **0.10**±0.05 | 0.26±0.22 | 0.21±0.10 | 0.89±0.18 | 0.40±0.06 |
| | | 70 | 0.68±0.33 | 0.11±0.02 | 0.25±0.25 | 0.19±0.10 | 0.85±0.17 | 0.31±0.05 |
| | | 100 | **0.65**±0.32 | **0.10**±0.02 | **0.23**±0.23 | **0.16**±0.09 | **0.83**±0.17 | **0.30**±0.02 |
| | Face | Base | 1.06±0.36 | 1.06±0.09 | 0.42±0.32 | 0.82±0.16 | 1.11±0.19 | 1.09±0.17 |
| | | 10 | 0.81±0.35 | 0.33±0.11 | 0.34±0.31 | 0.35±0.10 | 0.91±0.15 | 0.46±0.15 |
| | | 30* | 0.77±0.35 | 0.24±0.05 | 0.26±0.22 | 0.21±0.10 | 0.90±0.18 | 0.42±0.09 |
| | | 70 | 0.74±0.34 | 0.20±0.03 | 0.25±0.25 | 0.18±0.10 | 0.86±0.17 | 0.34±0.05 |
| | | 100 | **0.72**±0.34 | **0.19**±0.03 | **0.23**±0.23 | **0.17**±0.09 | **0.84**±0.17 | **0.31**±0.02 |

Table 2: Study of change in budget choice for the *far* g.t recourse scale. We show results on three datasets (in columns) and three baselines for the ANN classifier. We study the effect of incorporating user feedback using HARE with one iteration across a range of query budgets $B = 10, 30, 70, 100$. The results indicate that a budget of around $30 - 70$ strikes a balance between performance and cognitive load. The $\star$ indicates the default query budget used in Table 1.

**Study of Number of Iterations: Multi-HARE.** In Table 3, we present the results of Multi-HARE across varying number of iterations ($T = 1, 3, 5, 10$), showing the impact of iterative recourse refinement when keeping the budget fixed at $B = 30$. The results indicate that in general, increasing the number of iterations leads to generation of more personalized recourses. However performance tends to plateau or even degrade with more iterations. At higher iteration counts, the candidate set presented to the user at each round becomes smaller, which can restrict the diversity of recourses available for comparison. This can limit the extent to which user preferences can be incorporated. Across different baselines and datasets, $T = 5$ & $B = 30$ strikes a balance between performance and user cognitive load, and is a practical starting point for personalized recourse generation.

**Study of Choice of *BoundaryPointSearch*.** In § 3.4, we calibrate the initial candidate set by adjusting each candidate to move closer to the classifier's decision boundary, ensuring that the user's feature changes are smaller and thus easier to carry out. In this section we study the impact of calibrating only the best candidate instead of adjusting the entire candidate set. We solicit user feedback over the un-calibrated candidate set itself, to choose the best candidate. The *BoundaryPointSearch* algorithm is applied to calibrate only the best candidate. The results presented in Table 5 indicate that calibrating all candidates leads to better user satisfaction. Given that the binary search procedure incurs minimal computational overhead, calibrating all candidate recourses is an effective way of improving user satisfaction.

**Study of Ground-Truth Recourse Scale.** As described in § 4.1, we simulate ground-truth recourses at three scales – near, intermediate and far, to model users with different preferences. In Table 6, we present the performance of HARE and Multi-HARE for the ANN classifier. Both methods perform well across all scales, particularly in the *far* scenario, which is the most challenging in practice due to the larger magnitude of feature changes. Notably we observe significant gains in the GTD metric, indicating that our framework

| | Base Recourse | Iters (T) | ADULT | | COMPAS | | CREDIT | |
|---|---|---|---|---|---|---|---|---|
| | | | GTP ↓ | GTD ↓ | GTP ↓ | GTD ↓ | GTP ↓ | GTD ↓ |
| ANN | Wachter | 1 | 0.69±0.31 | 0.09±0.04 | 0.27±0.26 | 0.27±0.17 | 0.89±0.17 | 0.37±0.09 |
| | | 3 | **0.48**±0.23 | **0.07**±0.01 | **0.22**±0.18 | 0.22±0.14 | 0.89±0.19 | 0.36±0.09 |
| | | 5* | 0.58±0.28 | 0.08±0.03 | 0.27±0.26 | 0.26±0.15 | 0.90±0.22 | 0.38±0.12 |
| | | 10 | 0.59±0.29 | 0.10±0.03 | **0.22**±0.20 | **0.20**±0.14 | **0.87**±0.22 | **0.32**±0.04 |
| | GS | 1 | 0.71±0.34 | 0.10±0.05 | 0.26±0.22 | **0.21**±0.10 | 0.89±0.18 | 0.40±0.06 |
| | | 3 | **0.55**±0.25 | 0.10±0.02 | 0.27±0.26 | **0.21**±0.08 | 0.86±0.21 | 0.34±0.11 |
| | | 5* | 0.56±0.27 | 0.08±0.03 | 0.25±0.23 | 0.21±0.10 | **0.83**±0.21 | **0.30**±0.07 |
| | | 10 | 0.58±0.27 | **0.07**±0.02 | 0.28±0.28 | 0.23±0.10 | 0.85±0.17 | 0.33±0.05 |
| | Face | 1 | 0.77±0.35 | 0.24±0.05 | 0.26±0.22 | **0.21**±0.10 | 0.90±0.18 | 0.42±0.09 |
| | | 3 | **0.55**±0.26 | 0.11±0.03 | 0.27±0.26 | **0.21**±0.08 | 0.86±0.21 | 0.34±0.11 |
| | | 5* | 0.56±0.26 | **0.08**±0.02 | 0.25±0.24 | **0.21**±0.10 | **0.83**±0.22 | **0.32**±0.08 |
| | | 10 | 0.61±0.28 | **0.08**±0.03 | 0.27±0.28 | 0.22±0.09 | 0.86±0.18 | 0.34±0.03 |

Table 3: Study of change in iterations, Multi-HARE for the *far* g.t recourse scale. We show results on three datasets and three baselines for the ANN classifier. We study the effect of varying the number of iterations in Multi-HARE as $T = 1, 3, 5, 10$ with a fixed budget of $B = 30$. The results indicate that iterative refinement generates better personalized recourses. The $\star$ indicates the default number of iterations used in Table 1.

| Dataset | Base | Ours | GTP ↓ | GTD ↓ | Succ. Rate ↑ | Con. Vio. ↓ | Red. ↓ | Pro. ↓ | Spa. ↓ |
|---|---|---|---|---|---|---|---|---|---|
| ADULT | Wachter | Base | 0.93±0.38 | 0.87±0.03 | 1.00±0.00 | 0.00±0.00 | **3.30**±0.13 | **0.02**±0.00 | 4.00±0.00 |
| | | + HARE | 0.69±0.31 | 0.09±0.04 | 1.00±0.00 | 0.00±0.00 | 3.41±0.33 | 0.10±0.04 | 4.00±0.00 |
| | | + Multi-HARE | **0.58**±0.28 | **0.08**±0.03 | 1.00±0.00 | 0.00±0.00 | 3.46±0.14 | 0.25±0.12 | 4.00±0.00 |
| | GS | Base | 0.92±0.39 | 0.82±0.09 | 1.00±0.00 | 0.00±0.00 | **2.78**±0.06 | **0.01**±0.00 | 4.00±0.00 |
| | | + HARE | 0.71±0.34 | 0.10±0.05 | 1.00±0.00 | 0.00±0.00 | 3.43±0.26 | 0.09±0.02 | 4.00±0.00 |
| | | + Multi-HARE | **0.56**±0.27 | **0.08**±0.03 | 1.00±0.00 | 0.00±0.00 | 3.43±0.27 | 0.29±0.14 | 4.00±0.00 |
| | Face | Base | 1.06±0.36 | 1.06±0.09 | 1.00±0.00 | 2.01±0.02 | 2.80±0.32 | 0.20±0.02 | 4.22±0.13 |
| | | + HARE | 0.77±0.35 | 0.24±0.05 | 1.00±0.00 | 2.00±0.01 | 2.23±0.21 | **0.09**±0.03 | 6.04±0.02 |
| | | + Multi-HARE | **0.56**±0.26 | **0.08**±0.02 | 1.00±0.00 | 1.96±0.02 | 2.43±0.35 | 0.29±0.15 | 5.97±0.04 |
| COMPAS | Wachter | Base | 0.39±0.34 | 0.75±0.30 | 0.83±0.22 | 0.00±0.00 | 2.02±0.19 | **0.03**±0.02 | 2.95±0.01 |
| | | + HARE | **0.27**±0.26 | 0.27±0.17 | 0.82±0.22 | 0.00±0.00 | **1.67**±0.36 | 0.09±0.10 | 3.00±0.01 |
| | | + Multi-HARE | **0.27**±0.26 | **0.26**±0.15 | 0.82±0.22 | 0.00±0.00 | 1.74±0.32 | 0.09±0.10 | 2.99±0.01 |
| | GS | Base | 0.40±0.31 | 0.80±0.10 | 1.00±0.00 | 0.00±0.00 | **1.29**±0.12 | **0.04**±0.01 | 3.00±0.00 |
| | | + HARE | 0.26±0.22 | **0.21**±0.10 | 1.00±0.00 | 0.00±0.00 | 1.81±0.52 | 0.13±0.12 | 3.00±0.00 |
| | | + Multi-HARE | **0.25**±0.23 | **0.21**±0.10 | 1.00±0.00 | 0.00±0.00 | 1.53±0.46 | 0.15±0.16 | 3.00±0.00 |
| | Face | Base | 0.42±0.32 | 0.82±0.16 | 1.00±0.00 | 0.81±0.06 | **1.08**±0.05 | **0.06**±0.01 | 2.55±0.02 |
| | | + HARE | 0.26±0.22 | **0.21**±0.10 | 1.00±0.00 | 0.81±0.06 | 1.32±0.54 | 0.13±0.12 | 3.80±0.05 |
| | | + Multi-HARE | **0.25**±0.24 | **0.21**±0.10 | 1.00±0.00 | 0.75±0.10 | 1.12±0.53 | 0.14±0.14 | 3.70±0.09 |
| CREDIT | Wachter | Base | 1.07±0.18 | 0.97±0.26 | 0.86±0.18 | 0.00±0.00 | 4.99±0.21 | **0.03**±0.02 | 6.90±0.07 |
| | | + HARE | **0.89**±0.17 | **0.37**±0.09 | 0.86±0.18 | 0.00±0.00 | 4.64±0.29 | 0.15±0.03 | 7.99±0.01 |
| | | + Multi-HARE | 0.90±0.22 | 0.38±0.12 | 0.86±0.18 | 0.00±0.00 | 4.55±0.51 | 0.14±0.07 | 7.99±0.01 |
| | GS | Base | 1.06±0.19 | 1.06±0.09 | 1.00±0.00 | 0.00±0.00 | 5.61±0.40 | **0.02**±0.01 | 7.99±0.02 |
| | | + HARE | 0.89±0.18 | 0.40±0.06 | 1.00±0.00 | 0.00±0.00 | 4.48±0.35 | 0.15±0.05 | 8.00±0.00 |
| | | + Multi-HARE | **0.83**±0.21 | **0.30**±0.07 | 1.00±0.00 | 0.00±0.00 | 4.15±0.39 | 0.21±0.06 | 8.00±0.00 |
| | Face | Base | 1.11±0.19 | 1.09±0.17 | 1.00±0.00 | 0.89±0.03 | 4.93±0.11 | **0.12**±0.01 | 6.05±0.10 |
| | | + HARE | 0.90±0.18 | 0.42±0.09 | 1.00±0.00 | 0.89±0.03 | 3.82±0.15 | 0.15±0.06 | 8.88±0.03 |
| | | + Multi-HARE | **0.83**±0.22 | **0.32**±0.08 | 1.00±0.00 | 0.87±0.03 | **3.67**±0.40 | 0.23±0.08 | 8.85±0.04 |

Table 4: Results with classical recourse metrics for the *far* g.t recourse scale. We show results on the ANN classifier on three datasets and three baselines. We use a budget of $B = 30$ and use $T = 5$ iterations for Multi-HARE. Our framework has low values of GTP and GTD, but has a higher proximity score indicating that personalized recourses are not necessarily closest to the user's feature profile.

generates recourses that align strongly with the user's preferred direction of feature change. These results demonstrate that HARE and Multi-HARE are effective in incorporating a wide variety of user preferences.

**Other Metrics.** In Table 4, we report classical recourse metrics for HARE and Multi-HARE. Since our framework can be integrated with any recourse generator, our *Success Rate* remains unchanged from the baseline recourse generator. We observe that our generated recourses have a large *Proximity* score but a lower GPT and GTD score, indicating that user preferred recourses are not necessarily those closest to the user's feature profile. The generic proximity score is the Euclidean distance between the factual point and the generated recourse. This simple user-agnostic metric does not account for the preferences of different

| Base Recourse | Ours | BoundaryPointSearch @Best | @Every | ADULT GTP ↓ | GTD ↓ | COMPAS GTP ↓ | GTD ↓ | CREDIT GTP ↓ | GTD ↓ |
|---|---|---|---|---|---|---|---|---|---|
| Wachter | Base | - | - | 0.93±0.38 | 0.87±0.03 | 0.39±0.34 | 0.75±0.30 | 1.07±0.18 | 0.97±0.26 |
|  | + Multi-HARE | ✓ | ✗ | 0.64±0.29 | 0.11±0.03 | 0.33±0.25 | 0.54±0.31 | 0.95±0.19 | 0.47±0.09 |
|  |  | ✗ | ✓ | **0.58**±0.28 | **0.08**±0.03 | **0.27**±0.26 | **0.26**±0.15 | **0.90**±0.22 | **0.38**±0.12 |
| GS | Base | - | - | 0.92±0.39 | 0.82±0.09 | 0.40±0.31 | 0.80±0.10 | 1.06±0.19 | 1.06±0.09 |
|  | + Multi-HARE | ✓ | ✗ | 0.68±0.30 | 0.12±0.04 | 0.30±0.22 | 0.47±0.22 | 0.89±0.19 | 0.39±0.02 |
|  |  | ✗ | ✓ | **0.56**±0.27 | **0.08**±0.03 | **0.25**±0.23 | **0.21**±0.10 | **0.83**±0.21 | **0.30**±0.07 |
| Face | Base | - | - | 1.06±0.36 | 1.06±0.09 | 0.42±0.32 | 0.82±0.16 | 1.11±0.19 | 1.09±0.17 |
|  | + Multi-HARE | ✓ | ✗ | 0.70±0.29 | 0.15±0.07 | 0.31±0.23 | 0.50±0.26 | 0.90±0.19 | **0.40**±0.03 |
|  |  | ✗ | ✓ | **0.56**±0.26 | **0.08**±0.02 | **0.25**±0.24 | **0.21**±0.10 | **0.83**±0.22 | 0.32±0.08 |

Table 5: Study of choice of *BoundaryPointSearch* for the *far* g.t recourse scale. We show results on 3 datasets and 3 baselines on ANN classifier for Multi-HARE with $T = 5$ iterations. We study the impact of calibrating just the best candidate using *BoundaryPointSearch* at each iteration (indicated by @Best in this table). Results show that calibrating all candidates leads to better user satisfaction.

| Dataset | Base Recourse | Ours | Near GTP ↓ | GTD ↓ | Intermediate GTP ↓ | GTD ↓ | Far GTP ↓ | GTD ↓ |
|---|---|---|---|---|---|---|---|---|
| ADULT | Wachter | Base | 0.18±0.06 | 0.70±0.26 | 0.58±0.23 | 0.89±0.09 | 0.93±0.38 | 0.87±0.03 |
|  |  | + HARE | 0.08±0.02 | 0.09±0.02 | 0.36±0.17 | 0.10±0.02 | 0.69±0.31 | 0.09±0.04 |
|  |  | + Multi-HARE | **0.07**±0.01 | **0.06**±0.03 | **0.29**±0.13 | **0.06**±0.02 | **0.58**±0.28 | **0.08**±0.03 |
|  | GS | Base | 0.15±0.06 | 0.50±0.19 | 0.56±0.25 | 0.74±0.13 | 0.92±0.39 | 0.82±0.09 |
|  |  | + HARE | 0.09±0.04 | 0.12±0.07 | 0.39±0.17 | 0.10±0.03 | 0.71±0.34 | 0.10±0.05 |
|  |  | + Multi-HARE | **0.08**±0.03 | **0.09**±0.04 | **0.30**±0.13 | **0.08**±0.04 | **0.56**±0.27 | **0.08**±0.03 |
|  | Face | Base | 0.39±0.04 | 0.84±0.15 | 0.73±0.21 | 1.03±0.11 | 1.06±0.36 | 1.06±0.09 |
|  |  | + HARE | 0.12±0.03 | 0.23±0.02 | 0.44±0.19 | 0.22±0.04 | 0.77±0.35 | 0.24±0.05 |
|  |  | + Multi-HARE | **0.10**±0.01 | **0.15**±0.08 | **0.31**±0.13 | **0.09**±0.04 | **0.56**±0.26 | **0.08**±0.02 |
| COMPAS | Wachter | Base | 0.12±0.07 | 0.45±0.10 | 0.27±0.21 | 0.72±0.29 | 0.39±0.34 | 0.75±0.30 |
|  |  | + HARE | **0.05**±0.02 | 0.12±0.15 | **0.17**±0.13 | 0.26±0.15 | **0.27**±0.26 | 0.27±0.17 |
|  |  | + Multi-HARE | 0.06±0.02 | **0.11**±0.03 | **0.17**±0.13 | **0.25**±0.12 | **0.27**±0.26 | **0.26**±0.15 |
|  | GS | Base | 0.11±0.03 | 0.44±0.08 | 0.29±0.17 | 0.74±0.06 | 0.40±0.31 | 0.80±0.10 |
|  |  | + HARE | 0.07±0.02 | 0.12±0.03 | **0.16**±0.10 | **0.20**±0.09 | 0.26±0.22 | **0.21**±0.10 |
|  |  | + Multi-HARE | **0.05**±0.01 | **0.10**±0.04 | 0.16±0.11 | 0.20±0.07 | **0.25**±0.23 | 0.21±0.10 |
|  | Face | Base | 0.15±0.04 | 0.51±0.01 | 0.31±0.18 | 0.73±0.13 | 0.42±0.32 | 0.82±0.16 |
|  |  | + HARE | **0.06**±0.02 | **0.11**±0.04 | **0.16**±0.11 | 0.21±0.09 | 0.26±0.22 | **0.21**±0.10 |
|  |  | + Multi-HARE | **0.06**±0.01 | **0.10**±0.04 | 0.16±0.12 | **0.20**±0.07 | **0.25**±0.24 | 0.21±0.10 |
| CREDIT | Wachter | Base | 0.20±0.03 | 0.68±0.19 | 0.65±0.03 | 0.88±0.25 | 1.07±0.18 | 0.97±0.26 |
|  |  | + HARE | **0.15**±0.01 | 0.38±0.05 | **0.52**±0.03 | **0.36**±0.08 | **0.89**±0.17 | **0.37**±0.09 |
|  |  | + Multi-HARE | **0.15**±0.01 | **0.34**±0.02 | **0.52**±0.08 | 0.36±0.10 | 0.90±0.22 | 0.38±0.12 |
|  | GS | Base | 0.21±0.02 | 0.90±0.13 | 0.66±0.05 | 1.04±0.10 | 1.06±0.19 | 1.06±0.09 |
|  |  | + HARE | 0.17±0.02 | 0.43±0.05 | 0.53±0.04 | 0.39±0.06 | 0.89±0.18 | 0.40±0.06 |
|  |  | + Multi-HARE | **0.16**±0.01 | **0.39**±0.11 | **0.49**±0.06 | **0.30**±0.04 | **0.83**±0.21 | **0.30**±0.07 |
|  | Face | Base | 0.33±0.03 | 0.83±0.18 | 0.73±0.04 | 1.06±0.13 | 1.11±0.19 | 1.09±0.17 |
|  |  | + HARE | 0.17±0.01 | 0.41±0.07 | 0.53±0.04 | 0.42±0.08 | 0.90±0.18 | 0.42±0.09 |
|  |  | + Multi-HARE | **0.16**±0.01 | **0.36**±0.10 | **0.49**±0.06 | **0.31**±0.04 | **0.83**±0.22 | **0.32**±0.08 |

Table 6: Study of change in ground-truth distance. We study the effect of simulating the ground-truth recourse at three scales – near, intermediate, and far. We show results on three datasets and three baselines for both HARE and Multi-HARE with $B = 30$ & $T = 5$. Both methods show good performance across all scales, indicating their effectiveness in incorporating a wide variety of human feedback. We observe siginificant gains in the GTD metric, especially for the difficult *far* scenario.

users, and is not a good measure for the effort the user has to undertake in order to accomplish the recourse (since the "true" cost function of the user is unknown). For instance, in Figure 1 of the main paper, a user who strongly prefers to increase their level of income would be willing to choose a recourse with a higher L2 cost. For a concrete example from the ADULT dataset, we take a random user with the feature profile (excluding other features for brevity) of {'capital_gain': 0, 'hours': 35, 'capital_loss':0, 'marital_status': Non-Married, 'age': 54}. The simulated ground-truth recourse for this individual is {'capital_gain': 17184.48, 'hours': 2, 'capital_loss':4247.81, 'marital_status': Non-Married, 'age': 54} indicating that such an individual prefers to boost capital-gains i.e make

money from investments and assets while drastically reducing the number of hours worked. The base recourse generator (Wachter in this case) suggests a recourse of {'capital_gain': 6907.5, 'hours': 41.7, 'capital_loss':300.3, 'marital_status': Non-Married, 'age': 54} which is unsatisfactory for this user's strong preference for reducing hours and increasing income through capital gains. However, HARE suggests the recourse {'capital_gain': 12536.8, 'hours': 20.5, 'capital_loss':1020.72, 'marital_status': Non-Married, 'age': 54} which is a more reasonable suggestion for this user. HARE proposes a recourse that allows the user to reduce their work hours while allowing them to increase capital gains. This makes the recourse more practical and better aligned with the user's preferences, though it is far away from the user's original features in a euclidean sense.

## 5 Conclusions and Future Work

In this work, we address the important challenge of generating actionable explanations for a model through algorithmic recourse. In particular, we focus on personalizing algorithmic recourse through a human-in-the-loop framework that allows a user to choose appropriate counterfactuals that may work for them. To the best of our knowledge, this is a first effort on human-in-the-loop recourse that considers user preferences over counterfactuals. Our proposed framework, HARE, enhances traditional recourse generating algorithms by eliciting user preferences over candidate counterfactuals, which are used to generate better recourses. Our method can be used on top of any existing recourse generation methods, thus making it highly generalizable. Considering the limited efforts so far, we also propose an evaluation scheme to measure user satisfaction by simulating a ground-truth recourse. Our experimental evaluation shows that HARE generates recourses that significantly outperform earlier methods in terms of acceptance by end-users. HARE can generate better personalized recourses with as few as 10 comparison queries to the user. We hope that our framework can aid in the practical deployment of personalized recourse generation.

**Broader Impact and Limitations.** HARE terminates when the user is satisfied with a presented recourse, or when the user's budget is exhausted. HARE is flexible, allowing end users to dictate the kinds of choices available to them. For example, the ActionableSampling algorithm 1 ensures that candidates are diverse. It also allows users to specify new actionability constraints on the fly, and to restrict the range of sampling by tweaking $\gamma$. For consequential decision making systems (loans, jobs, crime etc), it is in the end user's best interest to achieve a recourse that they can act on, and our method presents a simple and modular framework to the end user, where user preference can be integrated with any existing recourse generation method. However, addressing the limitations of base recourse generation methods is beyond the scope of this work. Furthermore, care must be taken to ensure that the presented options do not bias the end-user into choosing an unfavorable recourse. This can be ensured by having a transparent candidate generation algorithm, and by having a mechanism where the end user can abstain from selecting any suggested candidate recourse. Further work is needed to ensure that personalization enhances end-user agency without introducing unfair or misleading recourses.

**Acknowledgements.** Sai Srinivas Kancheti would like to thank MoE for the PMRF fellowship support; he along with Vineeth N Balasubramanian, would also like to thank Microsoft India for the MAPG 2022 grant, under which this work was carried out. Rahul Vigneswaran would like to thank the Reliance Foundation for their Postgraduate Fellowship. We thank the anonymous reviewers for their valuable feedback that improved the presentation of this paper.

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
