## Appendix

In this appendix, we present the following additional details:

- In § A we compare our proposed approach HARE with PEAR Toni et al. (2024).

- § B has additional experiments and results

- In § C we present a brief description of baseline recourse generators

- In § D we present other hyperparameters

## A   Comparison with PEAR (Toni et al., 2024)

While both PEAR (Toni et al., 2024) and our method HARE are human-in-the-loop approaches for preference-based algorithmic recourse, we differ in the following important ways:

- **Different approaches to user preference-based algorithmic recourse**: PEAR assumes that the user's cost is parametrized according to a DAG with a weight $w$, and their method focuses on recovering these preferences. (Such an approach allows for validation of their solution with the given weight $w$ as ground truth, but does not generalize to other scenarios where a user's cost cannot be parametrized this way.) Their queries are not strictly over recourses but over a set of *interventions*. Each intervention is a sequence of actions from a start state (which may be different from the user's initial feature for $t > 1$) to the end state. PEAR 'discards' the intervention chosen at each time-step, using it just to update the estimates of $w^t$), which is then used to compute the final recourse given to the user. This is in contrast to HARE which takes preferences directly over the recourses. We allow the user to choose the terminating recourse directly from the candidate set. The benefit of our approach is that we make no assumptions whatsoever about the user's ground-truth cost, and incorporate user feedback in a more direct manner. While PEAR does perform preference-based algorithmic recourse, it is inherently a different approach, and we believe there is merit in both these approaches in their own ways. Our approach focuses on being general to any recourse-generator (unlike PEAR) and provides greater agency to the end user when acting directly at a recourse level.

- **Allow the user to choose their own path to the recourse**: PEAR provides a sequence of actions for the user to undertake. Continuing with the earlier point, this may be beneficial at times, but could also hamper the freedom of the end user to achieve their goal state. There is evidence Singh et al. (2023b) that users may prefer direct recourses for multiple reasons and providing a sequence of actions provides no additional benefit compared to simply providing the target recourse, as HARE does. Furthermore the sequences generated by PEAR depend on the causal relationships between features, which are difficult to predict in practice for a specific user. Incorrect assumptions in the causal DAG can result in sequential recourses that are impossible for the use to achieve in practice. (Note that we do not claim that PEAR is not useful, we only state that PEAR and HARE cater to different user application contexts.)

- **Generalizable with any recourse generator**: The recourse generator in PEAR is an extension of FARE Toni et al. (2023), which generates interventions, i.e. PEAR cannot be used with any recourse generator. In contrast, our method can be used on top of any recourse generating method. This was an important objective in our design, which we also show empirically in Tables 1- 4. On the other hand, PEAR is not designed to be integrated with any other recourse generator (such as Wachter, GS etc).

- **Flexibility**: Our method HARE is flexible on account of being simple, and many components can be tweaked easily to account for the preferences of the end user. For instance, we can allow the user to pick 2 best recourses out of the candidate set instead of just 1, and run ActionableSampling 1

separately for both chosen recourses. The ActionableSampling algorithm can also be modified on-the-fly to incorporate new actionability constraints of the user. A key advantage of HARE is that our framework itself is simple and directly flexible from an end-user's perspective, thus providing more control to the user.

In summary, while sharing the objectives of PEAR, our goal in proposing HARE is to have a simple, modular and easily generalizable approach for generating personalized recourses. It is often non-trivial to associate a form and nature to a user's cost function. HARE thus makes no assumptions on the user's cost, and relies only on the counterfactual choices made by the interacting user. HARE can work on top of any existing recourse generator and its empirical results provide strong evidence to show that such simple systems can work in practice.

# B   Additional Results

In this section, we provide details of additional analysis on HARE. We first present an alternative evaluation protocol, where we associate each user with a simple, linear cost function instead. We then study the impact of noisy user choices in evaluating comparison queries. Finally we present the results of HARE on very low query budgets.

## B.1   Simulating User Preferences via a Cost Function

HARE makes no assumptions on the parametric form of a user's underlying unknown cost function. While this is a practical assumption, it has the disadvantage of not having a specific ground truth to study our solution's effectiveness. We hence simulate a "ground-truth" recourse to simulate user feedback and to aid in empirical evaluation. However, to show that HARE can work for other evaluation schemes, we simulate a preferences vector over user features $w$ (values ranging from 0.1 to 5) and simulate a user's cost as

$$c(x(u), r|w(u)) = \sum_{i=1}^{d} w(u)_i \cdot |x(u)_i - r_i|$$

where $u$ is the user, $w(u)$ indicates their feature preferences, $x(u)$ is the user's initial feature and $r$ is the recourse. We now simulate the user's choice

$$r_i \succ_u r_j \qquad \text{if} \quad c(x(u), r_i|w(u)) < c(x(u), r_j|w(u))$$

Note that we cannot compare directly with the cost form used by PEAR as the latter requires sequential interventions, which we do not have in our method. The results of the above experiment are shown in Table 7 below for the ADULT and COMPAS datasets on the ANN classifier over three recourse methods. We use B=30 queries, and T=5 for Multi-HARE. Recall that *base* refers to the corresponding recourse generator's original performance before integration with HARE. Proximity is the L2 distance between the generated recourse and the original feature, and cost is the simulated cost with user preferences as described above. HARE and Multi-HARE have lower costs when compared to the base recourse method.

## B.2   Noisy User Feedback

In all our experiments we assume that the user feedback is noiseless. In this section we study the performance of HARE when user feedback is noisy. We study two noise models – i) A bernoulli noise model where for every comparison query the user picks the 'wrong' alternative (candidate further away from the simulated ground-truth) with probability $\eta$. ii) A more realistic logistic noise model. Given two counterfactual choices $r_i, r_j$, we model the probability of making an error as $1 - \frac{1}{1+e^{-\alpha \cdot d(r_i, r_j)+\beta}}$ where $\alpha, \beta$ are positive constants. When two recourses are indistinguishably close, the user has a greater chance of making a mistake, and when they are very far apart, the correct choice is clear and the user makes no error. We set $\alpha = 2, \beta = 1$ so that the noise level is 0.1 when d=0.

| | Base | Ours | ADULT | | COMPAS | | CREDIT | |
|---|---|---|---|---|---|---|---|---|
| | Recourse | | Cost ↓ | Proximity ↓ | Cost ↓ | Proximity ↓ | Cost ↓ | Proximity ↓ |
| ANN | Wachter | Base | 0.46±0.07 | 0.12±0.02 | 0.41 ± 0.23 | 0.14 ± 0.05 | 0.67 ± 0.17 | 0.13 ± 0.03 |
| | | + HARE | **0.22**±0.09 | **0.08**±0.02 | **0.23** ± 0.29 | 0.13 ± 0.06 | **0.34** ± 0.13 | **0.09** ± 0.04 |
| | | + Multi-HARE | 0.27±0.10 | 0.11±0.01 | 0.26 ± 0.28 | 0.15 ± 0.07 | 0.36 ± 0.11 | 0.10 ± 0.04 |
| | GS | Base | 0.24±0.10 | 0.09±0.01 | 0.39 ± 0.30 | 0.15 ± 0.00 | 0.38 ± 0.13 | 0.09 ± 0.03 |
| | | + HARE | **0.20**±0.10 | **0.08**±0.01 | 0.30 ± 0.28 | 0.15 ± 0.02 | 0.33 ± 0.11 | 0.09 ± 0.03 |
| | | + Multi-HARE | 0.21±0.09 | **0.08**±0.01 | **0.29** ± 0.28 | 0.16 ± 0.02 | **0.32** ± 0.11 | 0.10 ± 0.04 |
| | FACE | Base | 0.95±0.07 | 0.32±0.02 | 0.52 ± 0.39 | 0.19 ± 0.01 | 1.01 ± 0.26 | 0.30 ± 0.02 |
| | | + HARE | **0.27**±0.10 | **0.10**±0.02 | 0.29 ± 0.25 | 0.15 ± 0.02 | 0.41 ± 0.09 | **0.13** ± 0.02 |
| | | + Multi-HARE | 0.28±0.12 | 0.12±0.02 | **0.27** ± 0.25 | **0.15** ± 0.01 | **0.37** ± 0.13 | 0.13 ± 0.04 |
| LR | Wachter | Base | 1.00±0.02 | 0.26±0.00 | 0.41 ± 0.13 | **0.12** ± 0.01 | 0.78 ± 0.22 | **0.16** ± 0.00 |
| | | + HARE | 0.58±0.19 | **0.24**±0.02 | 0.15 ± 0.08 | 0.18 ± 0.05 | **0.56** ± 0.14 | 0.17 ± 0.01 |
| | | + Multi-HARE | **0.61**±0.20 | 0.27±0.02 | **0.13** ± 0.06 | 0.19 ± 0.04 | 0.63 ± 0.13 | 0.18 ± 0.02 |
| | GS | Base | 0.61±0.23 | 0.23±0.00 | 0.38 ± 0.26 | **0.13** ± 0.00 | 0.68 ± 0.11 | **0.16** ± 0.00 |
| | | + HARE | 0.52±0.22 | **0.22**±0.01 | 0.19 ± 0.12 | 0.14 ± 0.02 | **0.52** ± 0.11 | **0.14** ± 0.01 |
| | | + Multi-HARE | 0.53±0.23 | 0.23±0.01 | **0.15** ± 0.10 | 0.17 ± 0.04 | 0.59 ± 0.09 | 0.16 ± 0.02 |
| | Face | Base | 1.79±0.11 | 0.59±0.03 | 0.54 ± 0.39 | 0.21 ± 0.02 | 1.00 ± 0.21 | 0.31 ± 0.01 |
| | | + HARE | 0.76±0.15 | **0.26**±0.04 | 0.17 ± 0.10 | **0.14** ± 0.02 | **0.49** ± 0.14 | **0.16** ± 0.01 |
| | | + Multi-HARE | **0.65**±0.20 | 0.30±0.04 | **0.13** ± 0.08 | 0.17 ± 0.03 | 0.59 ± 0.14 | 0.19 ± 0.00 |

Table 7: Results on simulating a linear cost for users. We show results on three datasets (in columns), and three baseline recourse generators (in rows) for two classifiers, ANN & LR. For each baseline, we show the impact of incorporating user feedback using HARE and Multi-HARE. Results in **bold** indicate best numbers, and underline second-best. We observe that our framework can effectively incorporate user feedback even in this evaluation scheme.

| | Base | Ours | ADULT | | COMPAS | | CREDIT | |
|---|---|---|---|---|---|---|---|---|
| | Recourse | | GTP ↓ | GTD ↓ | GTP ↓ | GTD ↓ | GTP ↓ | GTD ↓ |
| ANN | Wachter | Base | 0.93±0.38 | 0.87±0.03 | 0.39±0.34 | 0.75±0.30 | 1.07±0.18 | 0.97±0.26 |
| | | + Multi-HARE(noiseless) | **0.58**±0.28 | **0.08**±0.03 | **0.27**±0.26 | **0.26**±0.15 | **0.90**±0.22 | **0.38**±0.12 |
| | | + Multi-HARE ($\eta$=0.01) | **0.58** ±0.28 | **0.08** ±0.03 | **0.27** ±0.26 | **0.26** ±0.15 | 0.94 ±0.20 | 0.46 ±0.11 |
| | | + Multi-HARE ($\eta$=0.05) | **0.58** ±0.27 | 0.10 ±0.02 | 0.33 ±0.29 | 0.42 ±0.32 | 0.96 ±0.20 | 0.51 ±0.13 |
| | | + Multi-HARE ($\eta$=0.1) | 0.65 ±0.16 | 0.24 ±0.27 | 0.35 ±0.28 | 0.49 ±0.29 | 0.99 ±0.17 | 0.64 ±0.04 |
| | | + Multi-HARE ($\eta$=0.2) | 0.68 ±0.18 | 0.26 ±0.26 | 0.39 ±0.29 | 0.65 ±0.29 | 1.03 ±0.20 | 0.79 ±0.14 |
| | GS | Base | 0.92±0.39 | 0.82±0.09 | 0.40±0.31 | 0.80±0.10 | 1.06±0.19 | 1.06±0.09 |
| | | + Multi-HARE(noiseless) | **0.56**±0.27 | **0.08**±0.03 | **0.25**±0.23 | **0.21**±0.10 | **0.83**±0.21 | **0.30**±0.07 |
| | | + Multi-HARE ($\eta$=0.01) | 0.57 ±0.27 | **0.08** ±0.03 | **0.25** ±0.23 | **0.21** ±0.10 | 0.85 ±0.19 | 0.33 ±0.06 |
| | | + Multi-HARE ($\eta$=0.05) | 0.60 ±0.27 | 0.09 ±0.02 | 0.26 ±0.25 | 0.22 ±0.11 | 0.94 ±0.16 | 0.49 ±0.08 |
| | | + Multi-HARE ($\eta$=0.1) | 0.70 ±0.23 | 0.29 ±0.25 | 0.28 ±0.23 | 0.30 ±0.15 | 1.02 ±0.21 | 0.72 ±0.19 |
| | | + Multi-HARE ($\eta$=0.2) | 0.77 ±0.27 | 0.39 ±0.26 | 0.29 ±0.24 | 0.35 ±0.18 | 1.05 ±0.19 | 0.81 ±0.17 |
| | FACE | Base | 1.06±0.36 | 1.06±0.09 | 0.42±0.32 | 0.82±0.16 | 1.11±0.19 | 1.09±0.17 |
| | | + Multi-HARE(noiseless) | **0.56**±0.26 | **0.08**±0.02 | **0.25**±0.24 | **0.21**±0.10 | **0.83**±0.22 | **0.32**±0.08 |
| | | + Multi-HARE ($\eta$=0.01) | 0.58 ±0.24 | 0.13 ±0.08 | 0.27 ±0.23 | 0.28 ±0.21 | 0.84 ±0.22 | 0.34 ±0.08 |
| | | + Multi-HARE ($\eta$=0.05) | 0.59 ±0.24 | 0.12 ±0.08 | 0.28 ±0.25 | 0.29 ±0.21 | 0.88 ±0.18 | 0.40 ±0.06 |
| | | + Multi-HARE ($\eta$=0.1) | 0.64 ±0.28 | 0.16 ±0.03 | 0.28 ±0.23 | 0.33 ±0.24 | 0.93 ±0.20 | 0.49 ±0.07 |
| | | + Multi-HARE ($\eta$=0.2) | 0.76 ±0.35 | 0.29 ±0.13 | 0.33 ±0.27 | 0.44 ±0.24 | 0.97 ±0.20 | 0.56 ±0.08 |

Table 8: Results for user feedback with bernoulli noise ($\eta = \{0.01, 0.05, 0.1, 0.2\}$). We show results on three datasets (in columns), and three baseline recourse generators (in rows) for the ANN classifier. Multi-HARE ($\eta$) indicates that every pairwise comparison the user makes has a chance of being incorrect with probability $\eta$. HARE is fairly robust to user noise, as it has better GTP&GTD scores compared to the baseline recourse generator.

Note that every comparison made by the user is noisy in this study. For budget $B = 30$ and $T = 5$, there are 6 candidates at each time-step. To choose the best recourse out of a set of 6 candidates, we randomly pick 2 candidates and compare them. Thus a noise level of 0.1 is also fairly high, and leads to a correct choice with only a probability of $\approx 0.74$ at each time step. The results shown in Tables 8& 9 indicate that our method is fairly robust to realistic noise levels. It is also straightforward to use existing algorithms on noisy comparisons Yue et al. (2012); Falahatgar et al. (2017) to help select the best recourse. We leave this exploration for future work.

| Base | Ours | ADULT | | COMPAS | | CREDIT | |
|------|------|-------|-------|--------|-------|--------|-------|
| Recourse | | GTP ↓ | GTD ↓ | GTP ↓ | GTD ↓ | GTP ↓ | GTD ↓ |
| | Wachter | Base | | | | | |
| | | 0.93±0.38 | 0.87±0.03 | 0.39±0.34 | 0.75±0.30 | 1.07±0.18 | 0.97±0.26 |
| | | + Multi-HARE(logistic) | | | | | |
| | | 0.66 ±0.31 | 0.11 ±0.02 | 0.43 ±0.29 | 0.73 ±0.38 | 1.07 ±0.19 | 0.53 ±0.26 |
| | | + Multi-HARE(noiseless) | | | | | |
| | | **0.58**±0.28 | **0.08**±0.03 | **0.27**±0.26 | **0.26**±0.15 | **0.90**±0.22 | **0.38**±0.12 |
| ANN | GS | Base | | | | | |
| | | 0.92±0.39 | 0.82±0.09 | 0.40±0.31 | 0.80±0.10 | 1.06±0.19 | 1.06±0.09 |
| | | + Multi-HARE(logistic) | | | | | |
| | | 0.63 ±0.23 | 0.25 ±0.22 | 0.26 ±0.21 | 0.26 ±0.10 | 1.02 ±0.17 | 0.48 ±0.14 |
| | | + Multi-HARE(noiseless) | | | | | |
| | | **0.56**±0.27 | **0.08**±0.03 | **0.25**±0.23 | **0.21**±0.10 | **0.83**±0.21 | **0.30**±0.07 |
| | FACE | Base | | | | | |
| | | 1.06±0.36 | 1.06±0.09 | 0.42±0.32 | 0.82±0.16 | 1.11±0.19 | 1.09±0.17 |
| | | + Multi-HARE(logistic) | | | | | |
| | | 0.68 ±0.32 | 0.23 ±0.12 | 0.28 ±0.19 | 0.39 ±0.23 | 1.06 ±0.12 | 0.58 ±0.04 |
| | | + Multi-HARE(noiseless) | | | | | |
| | | **0.56**±0.26 | **0.08**±0.02 | **0.25**±0.24 | **0.21**±0.10 | **0.83**±0.22 | **0.32**±0.08 |

Table 9: Results for user feedback with logistic noise ($\alpha = 2, \beta = 1$). We show results on three datasets (in columns), and three baseline recourse generators (in rows) for the ANN classifier. Multi-HARE (noisy) indicates that every pairwise comparison the user makes has a chance of being noisy. HARE is fairly robust to user noise, as it has better GTP&GTD scores compared to the baseline recourse generator.

| Base | Budget | ADULT | | COMPAS | | CREDIT | |
|------|--------|-------|-------|--------|-------|--------|-------|
| Recourse | (B) | GTP ↓ | GTD ↓ | GTP ↓ | GTD ↓ | GTP ↓ | GTD ↓ |
| | Base | 0.93±0.38 | 0.87±0.03 | 0.39±0.34 | 0.75±0.30 | 1.07±0.18 | 0.97±0.26 |
| | 30* | **0.69**±0.31 | **0.09**±0.04 | **0.27**±0.26 | **0.27**±0.17 | **0.89**±0.17 | **0.37**±0.09 |
| | 1 | 0.90 ±0.40 | 0.64 ±0.18 | 0.38 ±0.34 | 0.74 ±0.33 | 1.06 ±0.18 | 0.95 ±0.24 |
| | 2 | 0.85 ±0.35 | 0.42 ±0.14 | 0.28 ±0.18 | 0.37 ±0.16 | 1.04 ±0.19 | 0.82 ±0.17 |
| | 3 | 0.81 ±0.32 | 0.43 ±0.09 | 0.38 ±0.35 | 0.73 ±0.32 | 0.99 ±0.17 | 0.57 ±0.12 |
| Wachter | 4 | 0.83 ±0.34 | 0.36 ±0.09 | 0.35 ±0.30 | 0.40 ±0.09 | 1.00 ±0.19 | 0.57 ±0.10 |
| | 5 | 0.75 ±0.36 | 0.19 ±0.08 | 0.34 ±0.31 | 0.48 ±0.23 | 1.01 ±0.20 | 0.63 ±0.07 |
| | 6 | 0.77 ±0.31 | 0.21 ±0.12 | 0.33 ±0.32 | 0.34 ±0.19 | 0.97 ±0.18 | 0.49 ±0.16 |
| | 7 | 0.76 ±0.36 | 0.21 ±0.09 | 0.35 ±0.32 | 0.49 ±0.22 | 0.97 ±0.15 | 0.49 ±0.19 |
| | 8 | 0.79 ±0.32 | 0.25 ±0.17 | 0.34 ±0.32 | 0.38 ±0.15 | 0.97 ±0.19 | 0.45 ±0.14 |
| | 9 | 0.70 ±0.29 | 0.15 ±0.11 | 0.33 ±0.32 | 0.38 ±0.16 | 0.96 ±0.18 | 0.48 ±0.05 |

Table 10: Study of change in budget choice for the *far* g.t recourse scale. We show results on three datasets (in columns) for the Wachter base recourse on the ANN classifier. We study the effect of incorporating user feedback using HARE with one iteration across a range of query budgets $B = \{1, \dots, 9\}$. The results indicate that HARE obtains gains in GTP&GTD even for small user budgets. The ⋆ indicates the default query budget used in Table 1.

## B.3 Budget Ablation

We expand on the budget ablation presented in Table 2 in § 4.4 by showing results on HARE on even lower budgets. In Table 10 below we show results for lower budgets $B = 1, 2, ..9$ for the Wachter Base Recourse for the far g.t recourse scale.

We observe that even for a small number of queries, there is significant gain in GTP and GTD for all three datasets. For instance, for the ADULT dataset, for B=5 queries, we observe the GTP drops from 0.93 to 0.75 and the GTD drops from 0.87 to 0.19. The 'base' row shows results using just the baseline recourse (which does not incorporate any feedback). The second row shows results for B=30 (the default setting in the paper, shown for comparison). We observe that in general as the number of comparison queries increases, we get more personalized recourses.

## C Baseline Recourse Generation Methods: Brief Description

We briefly describe salient recourse generation methods that we apply HARE to below:

**Wachter (Wachter et al., 2017).** Wachter finds a minimum cost counterfactual by gradient descent. It assumes a cost function $c(x(u), .) : \mathcal{X} \to \mathbb{R}_+$ that quantifies the amount of effort the individual has to take

to make the changes recommended by the recourse, and solves the optimization problem:

$$\mathcal{M}^{wachter}(x(u)) = \underset{r \in \mathcal{A}(u)}{\arg\min} \quad l(f(r), 1) + \lambda \cdot c(x(u), r) \tag{3}$$

where $f$ is the scoring function of the base classifier, $c$ is the cost function usually chosen as $l_1$ or $l_2$ distance, and $l : \mathbb{R} \times \{\pm 1\} \to \mathbb{R}_+$ is a loss function. Mean-squared loss and binary-cross-entropy loss are popularly used loss functions. During optimization, the first term ensures that the recourse is positively classified, and the second term encourages recourses to have a low cost. $\lambda$ is a trade-off hyperparameter. Note that the cost function used by Wachter is fixed and cannot encode user preferences.

**Growing Spheres (GS) (Laugel et al., 2017).** Growing Spheres is a random sampling-based heuristic to generate a low $l_2$ cost recourse. It proceeds by randomly sampling recourses in a ball around the feature vector. The radius of the ball is decreased until a valid recourse can no longer be found. A black-box access of the underlying classifier is assumed. GS cannot directly incorporate user feedback during generation.

**Feasible and Actionable Counterfactual Explanations (FACE) (Poyiadzi et al., 2019).** This method generates actionable counterfactuals that respect the underlying data manifold. A counterfactual is considered actionable only if it is reachable from the factual point via a high-density path in the feature space. A similarity graph is constructed over the training data, with the edge weights encoding both the euclidean distance and density between the data-points. The shortest path first algorithm (Dijkstra's algorithm) is then employed to find the nearest counterfactual. FACE also does not provide a mechanism to encoder individual preferences.

**CCHVAE, CLUE and CRUDS.** Similar to FACE, these methods generate recourses that lie on the data manifold. CCHVAE uses a Variational Autoencoder (VAE) to learn latent representation of mutable features given immutable features. The recourse is generated by sampling around the latent of the factual point. CRUDS uses a Conditional Subspace VAE to learn a disentangled latent space relevant for classification, which can be perturbed to generate recourses.

The recourse methods described above present 3 distinct strategies among the considered methods in their scope and approach. Wachter is a gradient-based method, GS is a sampling-based heuristic while Face, CCHAVE generates manifold-aware recourses. Our method, HARE, is a human-in-the-loop formulation that can build on top on any recourse-generating baseline method.

## D   Other Hyperparameters

We herein describe the hyperparameters used for the baseline recourse generators and the classifiers.

**Classifiers:** We use a 3-layered neural network with ReLU activations and $[10, 5, 10]$ neurons in each layer. Both Logistic Regression and NN models are trained using gradient descent with an Adam optimizer. We use a learning-rate of $2e - 3$, a batch-size of $256$ and we train for 20 epochs. The resulting classifiers have good accuracy on all datasets.

For **Wachter**, we use the $L1$ distance as the cost function. We set $\lambda = 0.1$, and use binary cross-entropy as the loss. We use the Adam optimizer to perform gradient-descent for 2500 iterations, with a learning-rate of 0.01. For **Growing Spheres**, we use the default values suggested by Laugel et al. (2017). For **FACE**, we use a knn graph over 10% of the training data. We use $L1$ distance to determine the minimum cost recourse. For **CCHVAE, CLUE & CRUDS**, we use the hyperparameters as provided in CARLA Pawelczyk et al. (2021).