# OpenReview forum: "HARE: Human-in-the-Loop Algorithmic Recourse"
_TMLR — Accepted by TMLR_

### Review · Reviewer_uV7n · 2024-11-20

**Summary Of Contributions:**

The paper improves upon the human centric actionable recourse literature. The paper proposes a human in the loop approach to provide an iteratively customizable recourse for improved actionability. This eventually leads to improved human satisfaction. This is achieved using a two way channel between the adversely affected individual and the ML model. To achieve this, the authors considers preferences over the counterfactuals rather than the individual features.

**Audience:**

Yes

**Broader Impact Concerns:**

This study is focussed on improving user acceptance of ML models by developing a human in the loop approach to modify the models decision. According to me, there are no major concerns related to ethical implications of this work.

**Claims And Evidence:**

Yes

**Requested Changes:**

1. It would be interesting to see the results for a much lower budget (B= 1, 2, 3 ... 10). This would help understand the difficulty of obtaining a cheaper recourse from an ML model.
2. A quick discussion (on the alternative options) would be helpful when user is unable to accept any of the candidates presented.
3. The proximity scores seems to be affected with the proposed system. Can the authors give some examples of the recourses generated and elaborate on how they are better.

**Strengths And Weaknesses:**

### Strengths

* Paper is well written and easy to follow.
* The study is well placed within the literature and covers the existing works very well.
* Sufficient analysis with experiments are included using real world datasets.
* I appreciate the detailed experimentation section.

### Weaknesses
Listed in Requested Changes section below

---

> ### Author Response · Authors · 2025-01-12
> **Response to Reviewer uV7n**
>
> Thank you for your thoughtful comments. We respond to each question/concern below.
>
> > (REQUESTED CHANGES) It would be interesting to see the results for a much lower budget (B= 1, 2, 3 ... 10). This would help understand the difficulty of obtaining a cheaper recourse from an ML model.
>
> Thank you for the interesting suggestion. Table 2 in Section 4.4 of our main paper showed the results of varying the budget B. Following this suggestion, in the table below, we show results for lower budgets B=1,2,..9 for the Wachter Base Recourse for the *far* g.t recourse scale.
>
> |Budget (B) | ADULT | ADULT | COMPAS |COMPAS| CREDIT |CREDIT|
> |:-:|:-:|:-:|:-:|:-:|:-:|:-:|
> ||GTP↓|GTD↓|GTP↓|GTD↓|GTP↓| GTD↓|
> |base|0.93±0.38| 0.87±0.03| 0.39±0.34 |0.75±0.30 |1.07±0.18| 0.97±0.26|
> |1|0.9±0.4| 0.64±0.18| 0.38±0.34| 0.74±0.33| 1.1±0.18| 0.95±0.24|
> |2|0.85±0.35 | 0.42±0.14 | 0.28±0.18 | 0.37±0.16 | 1.04±0.19 | 0.82±0.17|
> |3|0.81±0.32 | 0.43±0.09 | 0.38±0.35 | 0.73±0.32 | 0.99±0.17 | 0.57±0.12|
> |4|0.83±0.34 | 0.36±0.09 | 0.35±0.30 | 0.40±0.09 | 1.00±0.19 | 0.57±0.10|
> |5|0.75±0.36 | 0.19±0.08 | 0.34±0.31 | 0.48±0.23 | 1.01±0.20 | 0.63±0.07|
> |6|0.77±0.31 | 0.21±0.12 | 0.33±0.32 | 0.34±0.19 | 0.97±0.18 | 0.49±0.16|
> |7|0.76±0.36 | 0.21±0.09 | 0.35±0.32 | 0.49±0.22 | 0.97±0.15 | 0.49±0.19|
> |8|0.79±0.32 | 0.25±0.17 | 0.34±0.32 | 0.38±0.15 | 0.97±0.19 | 0.45±0.14|
> |9|0.70±0.29 | 0.15±0.11 | 0.33±0.32 | 0.38±0.16 | 0.96±0.18 | 0.48±0.05|
> |30|0.69±0.31 |0.09±0.04 |0.27±0.26 |0.27±0.17 |0.89±0.17| 0.37±0.09|
>
> We observe that even for a small number of queries, there is significant gain in GTP and GTD for all three datasets. For instance, for the ADULT dataset, for B=5 queries, we observe the GTP drops from 0.93 to 0.75 and the GTD drops from 0.87 to 0.19. The 'base' row shows results using just the baseline recourse (which does not incorporate any feedback). The final row shows results for B=30 (the default setting in the paper, shown for comparison). We observe that in general as the number of comparison queries increases, we get more personalized recourses. **We added these results to Section B.3 of the appendix in the revised version.**
>
> > (REQUESTED CHANGES) A quick discussion (on the alternative options) would be helpful when user is unable to accept any of the candidates presented.
>
> In practice, there must be a two-way dialog between the user and the recourse generator. If the user is unable to accept any of the candidates after many rounds of feedback, the recourse generator must be able to accept suggestions from the user and check if they are valid. For instance, consider the example from Figure 1 of the main paper where the user is provided the final 3 options (income, credit_score) of {(110,25), (130,26), (145,21)}. Even if the user is unable to accept any of these, her exposure to the choices allows her to suggest a possible recourse that works. For instance, she may state that (100,25) is doable, and in this example is a valid recourse. Note that the user may have an apriori implicit preference of changing income over credit score (perhaps they want to be future proof), but the exposure to counterfactual choices (which are constrained by the underlying predictive model) can reveal different preferences. **We added this discussion to section 3.4 of the revision.**

---

> > ### Author Response · Authors · 2025-01-12
> > **(cont.) Response to Reviewer uV7n**
> >
> > > (REQUESTED CHANGES) The proximity scores seems to be affected with the proposed system. Can the authors give some examples of the recourses generated and elaborate on how they are better.
> >
> > The generic proximity score is the Euclidean distance between the factual point and the generated recourse. This simple user-agnostic metric does not account for the preferences of different users, and is not a good measure for the effort the user has to undertake in order to accomplish the recourse (since the "true" cost function of the user is unknown). For instance, in Figure 1 of the main paper, a user who strongly prefers to increase their level of income would be willing to choose a recourse with a higher L2 cost. For a concrete example from the ADULT dataset, we take a random user with the feature profile (excluding other features for brevity) of
> > |capital_gain|hours|capital_loss|marital_status|age|
> > |-|-|-|-|-|
> > |0|35|0|Non-Married|54|
> >
> > The simulated ground-truth recourse for this individual is
> > |capital_gain|hours|capital_loss|marital_status|age|
> > |-|-|-|-|-|
> > |17184.48|2|4247.81|Non-Married|54|
> >
> > Such an invidual prefers to boost capital-gains (0->17148) i.e make money from investments and assests while drastically reducing the number of hours worked (from 35->2). The base recourse generator suggests a recourse of:
> >
> > |capital_gain|hours|capital_loss|marital_status|age|
> > |-|-|-|-|-|
> > |6907.5|41.7|300.3|Non-Married|54|
> >
> > which is unsatisfactory for this particular user's strong preference for reducing hours and increasing income through capital gains. However, HARE suggests a recourse:
> >
> > |capital_gain|hours|capital_loss|marital_status|age|
> > |-|-|-|-|-|
> > |12536.8|20.5|1020.72|Non-Married|54|
> >
> > which is a more reasonable suggestion for this user. HARE's recourse allows the user to reduce their work hours while allowing them to increase capital gains. This makes the recourse more practical and better aligned with the user’s preferences, though it may be farther from the user's original features in an Euclidean sense (higher proximity score).
> >
> > As stated in the common response to all reviewers, we make no assumptions on the parametric form of a user's underlying unknown cost function and hence simulate a user's ground truth and compare with our proposed solution (we study with multiple such possibilities for a robust evaluation). In the common response (to all reviewers, please see above), we also show results where we simulate the user's cost to show the usefulness of our methodology. **We have added this discussion to Section 4.4 of the revision.**
> >
> > We'd be happy to discuss and answer any further queries/concerns you may have.

---

### Review · Reviewer_1uEz · 2024-12-05

**Summary Of Contributions:**

The paper presents HARE, a human-in-the-loop method to generate personalized recourse recommendations by eliciting user’s preferences from binary comparison between candidate recourses. The authors evaluated their approach with simulation studies on multiple real-world datasets using different recourse generators.

The paper does a good job of presenting the scenario, the solution and the evaluation. I also believe the topic is interesting and in scope with TMLR. However, there is room for improvement, for example by considering more realistic cost functions or user models. Moreover, it is not clear how HARE departs significantly from current solutions for personalized recourse.

**Audience:**

Yes

**Broader Impact Concerns:**

I do not have concerns regarding the work's broader impact.

**Claims And Evidence:**

Yes

**Requested Changes:**

**[Required]** Perform an ablation experiment by considering Algorithm 4 with a more realistic user’s model of the relative/absolute feedback (e.g., noisy user).

**[Required]** Perform an ablation experiment with a more realistic cost function (e.g., weighted distance metric $c(x, w) = w^{\top}x$, where $w$ describes the preference of a user).

**[Required]** Better clarify the difference between HARE and De Toni et al. (2024) and the novelty of the proposed approach.

**[Optional]** Compare HARE with Yetukuri et al., (2023) and De Toni et al., (2024) since they seem to be reasonable baselines.

**Strengths And Weaknesses:**

When describing the strengths and weaknesses of this approach, I am not able to quantify the novelty of the proposed technique with respect to the state-of-the-art, and the reasoning behind some design choices I feel are too abstract and not realistic.

- **[Differences with De Toni et al., (2024)]** I do not understand the qualitative and quantitative differences between HARE and the cited approach proposed by De Toni et al. (2024). The paper cited by the authors **does** provide a **human-in-the-loop** approach for algorithmic recourse. In practice, their “interventions” $\mathcal{I}$ are the actions the user needs to perform to obtain the counterfactual $x_{cf}$ from $x_{f}$. Thus, claiming it concerns “feature-level” preferences is incorrect since the cost is computed over the entire intervention. As far as my understanding goes, their approach could be used with any counterfactual generator and cost function $c$, since the contribution is the interactive preference elicitation procedure for recourse. Indeed, the same authors performed also an initial evaluation with real humans integrating their approach with a simpler cost function [1].
Thus, the TMLR author’s claim that _“[...] this is the first of such effort to propose a framework to incorporate user feedback over counterfactuals to generate recourse”_ could be an overstatement of the contribution.

- **[Ground-truth recourses]** In Section 4.1, the authors define the “ground truth” recourses based on their $\ell_2$ “distance” from the initial instance $x_f$. However, it feels somewhat unrealistic, since users usually do not have one preferred recourse, but might favour different counterfactuals.

- **[Robustness of Algorithm 4 to user’s noise]** Algorithm 4 does not seem robust to a realistic interaction with a human. What if the human is noisy, and it provides a wrong comparison between $(r_j, r_i)$? Then, Algorithm 4 will not converge to the optimal recourse. Since Algorithm 4 does not learn a parametrized function (e.g., a Bradley-Terry model) leveraging previous user’s answers, HARE seems very impractical to use.

**[Questions]**

- Could you better qualify the quantitative difference between your approach and De Toni et al., (2024)?
- Could you better explain your notion of the “cost” of achieving recourse? How do you simulate the user’s feedback CE($r_i$, $r_j$) practically?
- How did you simulate the user interaction with HARE in your experiments (e.g., the relative/absolute feedback)?
- Does HARE provide any guarantees that we are indeed offering the user the cheaper recourse option? Namely, do Algorithms 2 and 3 provide the set of options minimizing the regret at every turn $t$?
- Why by increasing the number of iterations T (Table 3) we do not get improvements in certain scenarios in GTP (e.g., Adult)?

**[Minor Nitpicking]**
- In the Abstract, I would change the phrase _“[...] we observe that our framework significantly increases human satisfaction”_ since the experiments are all synthetic and the metrics used do not really model a realistic “user satisfaction”.
- What is the meaning of line 6 in Algorithm 4? $CE_u$ is a function, while $r_j,r_i$ are recourses, so the set definition is not very rigorous.

[1] Seyedehdelaram, et al. "Preference elicitation in interactive and user-centered algorithmic recourse: an initial exploration." Proceedings of the 32nd ACM Conference on User Modeling, Adaptation and Personalization. 2024.

---

> ### Author Response · Authors · 2025-01-12
> **Response to Reviewer 1uEz**
>
> Thank you for your thoughtful and detailed comments. We respond to each individual question/concern below.
>
> > When describing the strengths and weaknesses of this approach, I am not able to quantify the novelty of the proposed technique ... are too abstract and not realistic.
>
> We provide a detailed comparison with De Toni et al (PEAR) and a justification of the necessity of our approach with our design choices in the Common Response to all reviewers (please see above). Specifically, we emphasize that: (i) our method is a different take at user preference-based algorithmic recourse as compared to PEAR; (ii) PEAR proposes a path to a recourse, while we provide just a recourse allowing the user to use a path of their choice; (iii) PEAR is built on the FARE (De Toni 2023) recourse generation method, while our method can be integrated with any recourse generation method; (iv) our method is a simple, flexible framework that directly incorporates user preferences over recourses. We make no assumptions on the user's underlying cost function (unlike PEAR which assumes this knowledge). We answer other queries in the responses below.
>
> > [Differences with De Toni et al., (2024)] I do not understand the qualitative and quantitative differences ...  could be an overstatement of the contribution.
> >(Question) Could you better qualify the quantitative difference between your approach and De Toni et al., (2024)?
> >(REQUESTED CHANGES) Better clarify the difference between HARE and De Toni et al. (2024) and the novelty of the proposed approach.
>
> Thank you for this question. We concur that both PEAR and our method, HARE, have the same objective; however, they approach this differently. We have updated our manuscript to clarify our claims, as briefed in the response above as well as the common response.
>
> Although both De Toni et al (PEAR) and our method HARE are human-in-the-loop methods for generating personalized recourses, we differ in many important ways. PEAR uses user preferences over interventions to update the estimate of a parameterized cost with weights $w$. In direct contrast, we take user feedback directly over recourses. HARE allows end users to directly pick a terminating recourse from the candidate set. This is markedly different from PEAR, where only one feature is updated after a round of user feedback; such an approach (while it has its own applications) may provide intermediate interventions that may not be of direct interest to the user. Our goal in proposing HARE was to have a simple, modular and easily generalizable approach for generating personalized recourses. PEAR assumes knowledge of a parametrized form of a user's cost funciton, however, it is often non-trivial to associate a form and nature to a user's cost function. HARE thus makes no assumptions on the user's cost, and relies only on the counterfactual choices made by the interacting user. HARE can thus work on top of any existing recourse generator while PEAR is built on one method, FARE. Our method's empirical results provide strong evidence to show that such simple systems can work in practice. A more detailed discussion is given in the Common Response, and has been added to **Section A of the Appendix in the revised paper.**
>
> > [Ground-truth recourses] In Section 4.1, the authors define the “ground truth” recourses based on their “distance” from the initial instance . However, it feels somewhat unrealistic, since users usually do not have one preferred recourse, but might favour different counterfactuals.
>
> We do not make any assumptions about the user's underlying ground-truth cost function (unlike earlier work). Such an approach, while practical, has the disadvantage of not having a specific ground truth to study our solution's effectiveness. We hence simulate a "ground-truth" (gt) recourse to simulate user feedback and to aid in empirical evaluation. However, as described in Section 4.1 of the paper, we simulate these ground-truths at varying distance scales to empirically show that our algorithm HARE works for different types of users. We choose a single gt recourse for ease of evaluation, but HARE works even when there are multiple ground truths. As discussed in the common response, HARE is flexible and can be modified easily to allow the user to pick multiple best recourses from a candidate set.
>
> > [Robustness of Algorithm 4 to user’s noise] Algorithm 4 does not ... impractical to use.
> >(REQUESTED CHANGES) Perform an ablation experiment by considering Algorithm 4 with a more realistic user’s model of the relative/absolute feedback (e.g., noisy user).
>
> Thank you for the suggestion. We ran an experiment where user feedback is noisy with a probability $\eta$, i.e. the probability that a user prefers the "wrong" alternative (candidate further away from the simulated ground-truth) is $\eta$. The results for the Adult dataset on the ANN classifier for B=30 and num iters T=5 for three recourse generators are presented in the table below.

---

> > ### Author Response · Authors · 2025-01-12
> > **(cont) Response to Reviewer 1uEz**
> >
> > |NoiseLevel (eta) | Wachter ADULT | Wachter ADULT | GS ADULT |GS ADULT| FACE ADULT |FACE ADULT|
> > |:-:|:-:|:-:|:-:|:-:|:-:|:-:|
> > ||GTP↓|GTD↓|GTP↓|GTD↓|GTP↓| GTD↓|
> > |0|0.58±0.28 |0.08±0.03| 0.56±0.27 | 0.08±0.03| 0.56±0.26 | 0.08±0.02|
> > |0.01 |0.58±0.28 | 0.08±0.03 | 0.57±0.27 | 0.08±0.03 | 0.58±0.24 | 0.13±0.08|
> > |0.05|0.58±0.27 | 0.10±0.02 | 0.60±0.27 | 0.09±0.02 | 0.59±0.24 | 0.12±0.08|
> > |0.1|0.65±0.16 | 0.24±0.27 | 0.70±0.23 | 0.29±0.25 | 0.64±0.28 | 0.16±0.03|
> > |0.2|0.68±0.18 | 0.26±0.26 | 0.77±0.27 | 0.39±0.26 | 0.76±0.35 | 0.29±0.13|
> > |base |0.93±0.38 |0.87±0.03| 0.92±0.39 | 0.82±0.09 | 1.06±0.36 |1.06±0.09 |
> >
> > Note that every comparison made by the user is noisy in this study. We use 6 candidates at each time-step. To choose the best recourse out of a set of 6 candidates, we randomly pick 2 candidates and compare them. Thus a noise level of 0.1 is also fairly high, and leads to a correct choice with only a probability of $\approx 0.74$ at each time step. The results above show that HARE performs well at various noise levels.
> >
> > We also study a more realistic logistic noise model. Given two counterfactual choices $r_i$, $r_j$, we model the probability of making an error as $1-\frac{1}{1+e^{-\alpha\cdot d(r_i,r_j) + \beta}}$ where $\alpha, \beta$ are positive constants. When two recourses are indistinguishably close, the user has a greater chance of making a mistake, and when they are very far apart, the correct choice is clear and the user makes no error. We set $\alpha=2, \beta=1$ so that the noise level is 0.1 when d=0. As discussed above, a noise level of 0.1 for every comparison is fairly high. The results for this setting are given in the table below for the ANN classifier:
> >
> > | Recourse Generator | Ours | ADULT GTP  | ADULT GTD  | COMPAS GTP  | COMPAS GTD         |
> > |---------------------|-----------------------|----------------------|-------------------|----------------------|--------------------|
> > | Wachter             | Base  | 0.93 ± 0.38          | 0.87 ± 0.03       | 0.39 ± 0.34  | 0.75 ± 0.30        |
> > |                     | +Multi-HARE (noisy)  | 0.66 ± 0.31          | 0.11 ± 0.02       | 0.43 ± 0.29   | 0.73 ± 0.38        |
> > |                     | +Multi-HARE (noiseless)| 0.58 ± 0.28       | 0.08 ± 0.03  | 0.27 ± 0.26    | 0.26 ± 0.15        |
> > | GS                  | Base  | 0.92 ± 0.39          | 0.82 ± 0.09       | 0.40 ± 0.31     | 0.80 ± 0.10        |
> > |                     | +Multi-HARE (noisy)  | 0.63 ± 0.23          | 0.25 ± 0.22   | 0.26 ± 0.21       | 0.26 ± 0.10        |
> > |                     | +Multi-HARE (noiseless)| 0.56 ± 0.27       | 0.08 ± 0.03  | 0.25 ± 0.23  | 0.21 ± 0.10        |
> > | FACE                | Base   | 1.06 ± 0.36          | 1.06 ± 0.09       | 0.42 ± 0.32    | 0.82 ± 0.16        |
> > |                     | +Multi-HARE (noisy)  | 0.68 ± 0.32          | 0.23 ± 0.12   | 0.28 ± 0.19  | 0.39 ± 0.23        |
> > |                     | +Multi-HARE (noiseless)| 0.56 ± 0.26       | 0.08 ± 0.02   | 0.25 ± 0.24   | 0.21 ± 0.10        |
> >
> > The above results indicate that our method is fairly robust to realistic noise levels. It is also straightforward to use existing algorithms on noisy comparisons (Yue06, falahatgar17) to help select the best recourse. We leave this exploration for future work. **We have added this discussion to Section B.2 in the Appendix of the revised version.**
> >
> > > (QUESTION) Could you better explain your notion of the “cost” of ... CE(r_i,r_j) practically?
> >
> > Since we do not make any assumptions about the user's underlying costs, we measure "cost" through the user satisfication in obtaining a suitable recourse. As stated earlier, this is measured by simulating a ground-truth recourse (at various scales) for every user and by measuring the distance between the recourse generated by HARE and the simulated gt via our GTP\&GTD metrics as described in Section 4.3. (As stated earlier, we also vary this gt, and study the effectiveness of our method for completeness.)
> >
> > > (QUESTION) How did you simulate ... (e.g., the relative/absolute feedback)?
> >
> > We simulate the feedback via a simulated ground truth recourse. The output of $CE(r_i, r_j)$ is simply $r_i$ if $d(r_{gt}, r_i) < d(r_{gt}, r_j)$ else $r_j$. In our original experiments, we assumed user feedback to be noiseless, which we have clarified in the revised version. We however perform well for realistic noise models as shown in Tables 1 \& 2 above.
> >
> > > (QUESTION) Does HARE provide ... minimizing the regret at every turn t?
> >
> > Since we use the simulated groud-truth recourse to measure the goodness of our generated recourse, our GTP\& GTD metrics already measure the regret. For noiseless user feedback, it is easy to see that regret cannot increase as the user always picks the best candidate. Providing formal guarantees for progressive decrease of regret over iterations under noisy user feedback would be an interesting direction for future extensions of our work.

---

> > > ### Author Response · Authors · 2025-01-12
> > > **(cont. to above) Reponse to Reviewer 1uEz**
> > >
> > > >(QUESTION) Why by increasing the number of iterations T (Table 3) we do not get improvements in certain scenarios in GTP (e.g., Adult)?
> > >
> > > This could be because the sampling is not fine enough to obtain candidates close to the ground-truth. In practice, for later iterations, we can restrict the range from which candidate recourses are sampled via the ActionableSampling algorithm by decreasing $\gamma$ (Algorithm 1 line 3). To ensure fairness of comparison in our studies, we fix this parameter across all recourse methods and ground-truth scales.
> > >
> > > >(MINOR) In the Abstract, ... a realistic “user satisfaction”.
> > >
> > > Thank you, we have changed the line to "framework performs significantly better on simulated user preferences."
> > >
> > > > (MINOR) What is the meaning of line 6 in Algorithm 4? ... rigorous.
> > >
> > > $CE(r_i,r_j)$ is +1 if user prefers $r_i$ over $r_j$ and zero otherwise as defined in Section 3.3. Thank you for bringing this typo to our attention. The set in Line 6 of algorithm 1 now evaluates to $r_j$ if $CE(r_i,r_j)=1$ (i.e user prefers $r_i$).
> > >
> > > > (REQUESTED CHANGES) Perform an ablation experiment with a more realistic cost function c(x,w)=w^t x(e.g., weighted distance metric , where describes the preference of a user).
> > >
> > > We perform an experiment by considering a cost function of the form $c(x(u), r | w(u)) = \sum_{i=1}^d w(u)_i\cdot |x(u)_i - r_i|$ parametrized by a weight $w$. The detailed results are provided in point 2 of the Common Response (we don't repeat it here in the interest of space).
> > >
> > > > (REQUESTED CHANGES)  Compare HARE with Yetukuri et al., (2023) and De Toni et al., (2024) since they seem to be reasonable baselines.
> > >
> > > As stated in our responses, our approach and hence evaluation scheme differs significantly from PEAR (De Toni et al 2024). PEAR considers sequential interventions and thus cannot incorporate any recourse generating method other than FARE. Furthermore they assume that every user has a fixed form of a cost function, which is a very strong assumption that may not hold in practice. In contrast our evaluation scheme simulates ground-truth recourses for every user. Our method HARE works on top of any recourse generator while PEAR can only use FARE. As shown above, HARE works even when we assume some fixed form of user cost function. We do not compare with Yetukuri et al 2023 as they require an apriori vector of preferences over features and their method cannot be extended directly to incorporate a human-in-the-loop.
> > >
> > > To ensure consistency, we use the CARLA library for all base recourse generators in our setup (for purposes of standardization and alignment with progress in the community). Unfortunately, PEAR and FARE are not compatible with CARLA. While we attempted to run the FARE code provided by its authors, it did not work within the CARLA framework. However, we see no conceptual reason why HARE could not work with FARE as a base recourse generator.

---

### Review · Reviewer_Y1VJ · 2024-12-06

**Summary Of Contributions:**

The paper presents a method for personalized algorithm recourse, where low-cost, peronsalized propositions for feature value changes are recommended in order to change the output of a black-box classifier.
The authors generate possible recourse options from exisiting, re-usable recourse recommenders and use preference elicitation to learn which option is preferred by the end-user. The options are chosen to be maximally diverse and the preferred option is finally calibrated to be of minimal cost.
The approach is evaluated against several agnostic baselines, where the proposed approach is put on top. The chosen datasets are Adult Income, GiveMeSomeCredit, COMPAS. Preferences are artificially sampled, where the authors distinguish between differences distances.
The results show that the proposed approach generates more personalized recourses compared to the non-personalized baselines.

**Audience:**

Yes

**Broader Impact Concerns:**

- It should be discussed what biases personalized recourses might have on the end-user. By presenting multiple options during the interactions, one could manipulate the end-user.

**Claims And Evidence:**

No

**Requested Changes:**

- Discuss how it can be ensured that the preference elicitation loop converges. This might not happen, even if options are diverse. What if the base algorithms are not good enough for some cases?
- Argue why one does not have to evaluate against the personalized competitors or perform empirical comparison
- Dataset and personalization of evaluation differs from related work - what is the motivation for it?
- Please better discuss the differences to Toni et al. 2024 - it may be that the preference elicitation step is also conducted based on options in their approach.

**Strengths And Weaknesses:**

# Strenghts
- Personalization of algorithm recourse is fruitful, important topic. The proposed approach is sensible in that preference elicitation is used to find the best match
- Evaluation is done with numerous non-agnostic approaches, showing that method for personalization works well throughout

# Weaknesses
- Main claim for delta to related work (Toni et al. 2024) not clear. It is not clearly discussed how the competitor works. While feature interactions are modeled, the paper also proposes to have end-users select the best candidate option in multiple rounds.
- Main competitors (Toni et al. 2024 and Yetukuri et al. (2023)) are not compared against in empirical evaluation. If one does not give explicit reasons why a comparison is not needed / possible, one should evaluate against them.
- Dataset and personalization of evaluation differs from related work. While datasets overlap, it is not argued why one does not exactly replicate prior evaluations - also wrt to the personalization simulation.

---

> ### Author Response · Authors · 2025-01-12
> **Response to Reviewer Y1VJ**
>
> Thank you for your thoughtful comments. We respond to each question/concern below.
>
> > (WEAKNESSES) Main claim for delta to related work (Toni et al. 2024) not clear. It is not clearly discussed how the competitor works. While feature interactions are modeled, the paper also proposes to have end-users select the best candidate option in multiple rounds.
> > (REQUESTED CHANGES) Please better discuss the differences to Toni et al. 2024 - it may be that the preference elicitation step is also conducted based on options in their approach.
>
> We have provided a detailed discussion of the differences with De Toni et al, 2024 (PEAR) in the common response to all reviewers (above).
>
> Although both De Toni et al (PEAR) and our method HARE are human-in-the-loop methods for generating personalized recourses, we differ in many important ways. PEAR uses user preferences over interventions to update the estimate of a parameterized cost with weights $w$. In direct contrast, we take user feedback directly over recourses. HARE allows end users to directly pick a terminating recourse from the candidate set. This is markedly different from PEAR, where only one feature is updated after a round of user feedback; such an approach (while it has its own applications) may provide intermediate interventions that may not be of direct interest to the user. Our goal in proposing HARE was to have a simple, modular and easily generalizable approach for generating personalized recourses. PEAR assumes knowledge of a parametrized form of a user's cost funciton, however, it is often non-trivial to associate a form and nature to a user's cost function. HARE thus makes no assumptions on the user's cost, and relies only on the counterfactual choices made by the interacting user. HARE can thus work on top of any existing recourse generator while PEAR is built on one method, FARE (De Toni 2023). Our method's empirical results provide strong evidence to show that such simple systems can work in practice. A more detailed discussion is given in the Common Response, and **has been added to Section A of the Appendix in the revised paper.**
>
> > (WEAKNESSES) Main competitors (Toni et al. 2024 and Yetukuri et al. (2023)) are not compared against in empirical evaluation. If one does not give explicit reasons why a comparison is not needed / possible, one should evaluate against them.
> > (REQUESTED CHANGES) Argue why one does not have to evaluate against the personalized competitors or perform empirical comparison
>
> As stated in our responses, our approach and hence evaluation scheme differs significantly from PEAR (De Toni et al 2024). PEAR considers sequential interventions and thus cannot incorporate any recourse generating method other than FARE. Furthermore they assume that every user has a fixed form of a cost function, which is a very strong assumption that may not hold in practice. In contrast our evaluation scheme simulates ground-truth recourses for every user. Our method HARE works on top of any recourse generator while PEAR can only use FARE. As shown above, HARE works even when we assume some fixed form of user cost function. We do not compare with Yetukuri et al 2023 as they require an apriori vector of preferences over features and their method cannot be extended directly to incorporate a human-in-the-loop.
>
> > (WEAKNESSES) Dataset and personalization of evaluation differs from related work. While datasets overlap, it is not argued why one does not exactly replicate prior evaluations - also wrt to the personalization simulation.
> > (REQUESTED CHANGES) Dataset and personalization of evaluation differs from related work - what is the motivation for it?
>
> We do not assume a particular form of user cost and the lack of flexibility of PEAR makes direct empirical comparison challenging. In contrast, our evaluation scheme simulates ground-truth recourses for every user. We simulate a "ground-truth" (gt) recourse to simulate user feedback and to aid in empirical evaluation. We measure the goodness of the generated recourse by measuring its closeness to the ground-truth. As such, we report two measures of user satisfaction GTP\&GTD (described in Section 4.1 of our paper) along with classical recourse metrics like Proximity and Success Rate.
>
> To ensure consistency, we use the CARLA library for all base recourse generators in our setup (for purposes of standardization and alignment with progress in the community). Unfortunately, PEAR and FARE are not compatible with CARLA. While we attempted to run the FARE code provided by its authors, it did not work within the CARLA framework. Hence, differences in dataset preprocessing and test point selection make exact replication of their results difficult. However, we see no conceptual reason why HARE could not work with FARE as a base recourse generator.

---

> > ### Author Response · Authors · 2025-01-12
> > **(cont.) Response to Reviewer Y1VJ**
> >
> > >(REQUESTED CHANGES) Discuss how it can be ensured that the preference elicitation loop converges. This might not happen, even if options are diverse. What if the base algorithms are not good enough for some cases?
> >
> > HARE terminates when the user is satisfied with a presented recourse, or when the user's budget is exhausted. HARE is flexible, allowing end users to dictate the kinds of choices available to them. For example, our ActionableSampling algorithm ensures that candidates are diverse. It also allows users to specify new actionability constraints on the fly, and to restrict the range of sampling by tweaking $\gamma$. For consequential decision making systems (loans, jobs, crime etc), it is in the end user's best interest to achieve a recourse that they can act on, and our method presents a simple and modular framework to the end user (where the user preference can be integrated with any existing recourse generation method). As you may appreciate, addressing the limitations of base recourse generation methods is not our objective (we seek to introduce user preferences that can be integrated into these methods) and hence beyond the scope of this work.

---

### Author Response · Authors · 2025-01-12
**Common Response to the Reviewers**

We thank all reviewers for their thoughtful feedback. We are pleased to see the following encouraging comments from the reviewers:
1. The proposed method is well-motivated and practical (Y1VJ)
2. The topic is interesting and has practical importance (Y1VJ, 1uEz)
3. Empirical evaluation is comprehensive (uV7n)

As a summary, our proposed method HARE is a simple, modular and user-friendly approach to generate personalized recourses, that takes a different approach from existing works such as PEAR (De Toni el al 2024). It focuses on providing control to the end user. We believe that a venue like TMLR that encourages new ideas is an ideal venue for this work.

At the outset, we clarify two common points in the reviews below:
1. *Comparison with De Toni et al 2024 (PEAR):* We concur that PEAR and our method HARE consider user preferences over recourses using a human-in-the-loop approach. However, our method HARE takes a different approach and differs in the following important ways (we have revised the manuscript to clarify this too):
- **Different approaches to user preference-based algorithmic recourse**: De Toni et al 2024 (PEAR) assumes that the user's cost is parametrized according to a DAG with a weight $w$, and their method focuses on recovering these preferences. (Such an approach allows for validation of their solution with the given weight $w$ as ground truth, but does not generalize to other scenarios where a user's cost cannot be parametrized this way.) Their queries are not strictly over recourses but over a set of *interventions*. Each intervention is a sequence of actions from a start state (which may be different from the user's initial feature for t>1) to the end state. PEAR 'discards' the intervention chosen at each time-step, using it just to update the estimates of w^(t), which is then used to compute the final recourse given to the user. This is in contrast to HARE which takes preferences directly over the recourses. We allow the user to choose the terminating recourse directly from the candidate set. The benefit of our approach is that we make no assumptions whatsoever about the user's ground-truth cost, and incorporate user feedback in a more direct manner. While both PEAR and HARE perform preference-based algorithmic recourse, both are inherently different approaches, and we believe there is merit in both these approaches in their own ways. Our approach focuses on being general to any recourse-generator (unlike PEAR) and on providing greater agency to the end user by acting directly at a recourse level.
- **Allow user to choose their own path to the recourse**: PEAR provides a sequence of actions for the user to undertake. Continuing with the earlier point, this may be beneficial at times, but could also hamper the freedom of the end user to achieve their goal state. There is evidence (Ronal Singh et al 2021) that users may prefer direct recourses for multiple reasons and providing a sequence of actions provides no additional benefit compared to simply providing the target recourse, as HARE does. Furthermore the sequences generated by PEAR depend on the *causal* relationships between features, which are difficult to predict in practice for a specific user. Incorrect assumptions in the causal DAG can result in sequential recourses that are impossible for the use to achieve in practice. (Note that we do not claim that PEAR is not useful, we only state that PEAR and HARE cater to different user application contexts.)
- **Generalizable with any recourse generator**: The recourse generator in PEAR is an extension of FARE (De Toni 2023), which generates interventions, i.e. PEAR cannot be used with any recourse generator. In contrast, our method can be used on top of any recourse generating method. This was an important objective in our design, which we also show empirically (Tables 1,4 in our paper). On the other hand, PEAR is not designed to be integrated with any other recourse generator (such as Wachter, GS etc).
- **Flexibility**: Our method HARE is flexible on account of being simple, and its components can be tweaked easily to account for the preferences of the end user. For instance, we can allow the user to pick 2 best recourses out of the candidate set instead of just 1, and run ActionableSampling (Algorithm 1 of our paper) separately for both chosen recourses. The ActionableSampling algorithm can also be modified on-the-fly to incorporate new actionability constraints of the user. We expand on this discussion in Section 3.4 in the revision. A key advantage of HARE is that our framework itself is simple and directly flexible from an end-user's perspective, thus providing more control to the user.

---

> ### Author Response · Authors · 2025-01-12
> **(cont.) Common Response to the Reviewers**
>
> In summary, while sharing the objectives of De Toni et al (PEAR), our goal in proposing HARE was to have a simple, modular and easily generalizable approach for generating personalized recourses. It is often non-trivial to associate a form and nature to a user's cost function. HARE thus makes no assumptions on the user's cost, and relies only on the counterfactual choices made by the interacting user. HARE can work on top of any existing recourse generator and its empirical results provide strong evidence to show that such simple systems can work in practice. __We have added this discussion to Section A of the appendix in the revised version__.
>
> 2. *Our evaluation protocol by simulating a ground-truth recourse:* As stated in the above discussion, we  make no assumptions on the parametric form of a user's underlying unknown cost function. While this is a practical assumption, it has the disadvantage of not having a specific ground truth to study our solution's effectiveness. We hence simulate a "ground-truth" recourse to simulate user feedback and to aid in empirical evaluation. However, to show that HARE can work for other evaluation schemes, we simulate a preferences vector over user features $w$ (values ranging from 0.1 to 3) and simulate a user's cost as: $$c(x(u), r | w(u)) = \sum_{i=1}^d w(u)_i\cdot |x(u)_i - r_i|$$ where $u$ is the user, $w(u)$ indicate their feature preferences, $x(u)$ is the user's initial feature and $r$ is the recourse. We now simulate the user's choice as: $$r_i \succ_u r_j \qquad\text{if }\quad c(x(u), r_i | w(u)) < c(x(u), r_j | w(u))$$ Note that we cannot compare directly with the cost form used by PEAR as the latter requires sequential interventions, which we do not have in our method. The results of the above experiment are shown in the table below for the ADULT and COMPAS datasets on the ANN classifier over three recourse methods. We use B=30 queries, and T=5 for Multi-HARE. Recall that, as in our paper, *base* refers to the corresponding recourse generator's original performance before integration with HARE.
>
> | Recourse Generator | Ours                  | ADULT Cost         | ADULT Proximity   | COMPAS Cost        | COMPAS Proximity   |
> |---------------------|-----------------------|---------------------|-------------------|---------------------|--------------------|
> | Wachter             | base                 | 0.46 ± 0.07         | 0.12 ± 0.02       | 0.41 ± 0.23         | 0.14 ± 0.05        |
> |                     | +HARE                | 0.22 ± 0.09         | 0.08 ± 0.02       | 0.23 ± 0.29         | 0.13 ± 0.06        |
> |                     | +Multi-HARE          | 0.27 ± 0.10         | 0.11 ± 0.01       | 0.26 ± 0.28         | 0.15 ± 0.07        |
> | GS                  | base                 | 0.24 ± 0.10         | 0.09 ± 0.01       | 0.39 ± 0.30         | 0.15 ± 0.00        |
> |                     | +HARE                | 0.20 ± 0.10         | 0.08 ± 0.01       | 0.30 ± 0.28         | 0.15 ± 0.02        |
> |                     | +Multi-HARE          | 0.21 ± 0.09         | 0.08 ± 0.01       | 0.29 ± 0.28         | 0.16 ± 0.02        |
> | FACE                | base                 | 0.95 ± 0.07         | 0.32 ± 0.02       | 0.52 ± 0.39         | 0.19 ± 0.01        |
> |                     | +HARE                | 0.27 ± 0.10         | 0.10 ± 0.02       | 0.29 ± 0.25         | 0.15 ± 0.02        |
> |                     | +Multi-HARE          | 0.28 ± 0.12         | 0.12 ± 0.02       | 0.27 ± 0.25         | 0.15 ± 0.01        |
>
>
> Proximity is the L2 distance between the generated recourse and the original feature, and cost is the simulated cost with user preferences as described above. HARE has a lower cost when compared to the base recourse method. **We have added the full table to Section B.1 of the appendix in the revised version**.
>
> To summarize, HARE makes no assumptions on the user's cost, and relies only on the counterfactual choices made by the interacting user. HARE can work on top of any existing recourse generator and its empirical results provide strong evidence to show that such simple systems can work in practice. We take a different approach from existing works such as PEAR (De Toni el al 2024), and focus on providing more control to the end user.
>
> We once again thank reviewers for the thoughtful comments that helped us improve the presentation of our work. We have highlighted the changes in our revised manuscript in red for convenience of the reviewers. As stated earlier, we believe that a venue like TMLR that encourages different perspectives is well-suited for this work. We are happy to discuss further if required.

---

### Decision · Action_Editor_z9vd · 2025-02-07

**Recommendation:** Accept with minor revision

**Comment:**

After the author rebuttal and revision, all reviewers were in favor of acceptance. It is also my view that their main concerns noted above were well addressed.

I recommend a minor revision for the camera-ready version to address the following remaining points:
- Reviewer 1uEz brought up a recent work [1] after the rebuttal, whose title is clearly similar to this submission's and should be discussed in the paper.
- I think that Appendix A in the revision makes clear the distinction from PEAR (Toni et al., 2024). However, in my reading, the last paragraph of Section 1 is not as clear regarding the assumption by Toni et al. (2024) of a particular form for the user's cost. I think this is an important point that should also be clear in the main paper, specifically Section 1.
- The authors should consider responding to Reviewer Y1VJ's last comment about possible user manipulation.

[1] Abrate, Carlo, et al. "Human-in-the-loop personalized counterfactual recourse." World Conference on Explainable Artificial Intelligence. Cham: Springer Nature Switzerland, 2024.

**Audience:**

The reviewers agree that personalization of algorithmic recourse is an important topic, since the eventual goal is to have a user act upon the recommended recourses.

**Claims And Evidence:**

This submission describes a method called HARE for iteratively personalizing algorithmic recourses based on preference feedback from the user. The experiments show that the method can be applied on top of multiple recourse generation methods and significantly improve performance on simulated user preferences (the simulated aspect was clarified in the rebuttal and revised manuscript).

A common concern among the reviewers was the distinction from the PEAR method (De Toni et al., 2024), which also involves interaction with the user. The rebuttal and revision addressed this point to the reviewers' satisfaction. In brief, HARE does not assume a form for the user's cost function and thus can work with multiple recourse generation methods, whereas PEAR does assume a form for the cost and is thus restricted. HARE operates at the level of full recourses while PEAR works with single-feature interventions. These differences also complicate empirical comparison between the two.

Reviewer 1uEz had concerns related to the above claimed flexibility to different user cost models. The rebuttal and revision addressed this with two additional experiments, one using a different cost function and one including noise.

---

> ### Author Response · Authors · 2025-03-12
> **Camera-Ready Submission**
>
> We once again thank all reviewers and the AE for the constructive feedback, which has helped improve our paper. As suggested by the AE, we have compared against HIP-CORE[1] in Section 2, which we state here for completion. HIP-CORE solves a multi-objective problem which aims to generate a set of recourses that satisfy various metrics such as sparsity, diversity and user preference.  We differ from HIP-CORE in many important ways -- i) HARE can build on top of any recourse generator baseline to generate personalized recourses; ii) HIP-CORE elicits a numerical user-preference for every counterfactual leading to a huge cognitive burden for the user (they ask over $300$ numerical preferences per user). In contrast, we ask a small number of comparison queries (controlled via a budget $B=30$), making our approach practical; iii) HIP-CORE models preferences via a separate density function for each feature, whose parameters are learned using preferences. In contrast, we work directly with the user's preferred counterfactual; iv) Actionability constraints cannot be encoded into HIP-CORE's optimization; and finally v) HIP-CORE does not compare against many baselines such as Wachter, GS and FACE. We also note a significant issue with HIP-CORE's evaluation, where the true user preferences are simulated with the same family of densities that are considered by HIP-CORE's optimization, leading to unintended information leakage. In contrast, we simulate the ground-truth for every user in a non-parametric fashion.
>
> We have also modified the Introduction to make the distinction from PEAR clearer, and have added a Broader-Impacts and Limitations paragraph, wherein we discuss possible user manipulation. We release our code for reproducibility.
>
> Besides this, we state here the changes made in earlier versions, for completeness:
>
> - We have added various clarifications in response to reviewer comments in Sections 3.4 & 4.4
> - We perform a detailed comparison with PEAR, which we add to the main paper Sections 1 & 2 as well as Section A of the Appendix
> - We add a new "Random" baseline to Table 1 in the main paper
> - In Section B.1 of the Appendix, we study an alternate evaluation protocol where we simulate a ground-truth cost for each user, and show that HARE performs well even in this setting
> - In Section B.2 of the Appendix, we study the performance of HARE when user feedback is noisy. We consider two noise models -- Bernoulli and Logistic, and show that HARE is fairly robust to realistic noise levels
> - In Section B.3 of the Appendix, we study performance of HARE under very low budgets
>
> [1] Abrate, Carlo, et al. "Human-in-the-loop personalized counterfactual recourse." World Conference on Explainable Artificial Intelligence. Cham: Springer Nature Switzerland, 2024.